

# Quantifying the UK's Carbon Dioxide Flux: An atmospheric inverse modelling approach using a regional measurement network

Emily D. White[1], Matthew Rigby[1], Mark F. Lunt[1,2] Anita L. Ganesan[3], Alistair J. Manning[4], Simon O'Doherty[1], Ann R. Stavert[1,5], Kieran M. Stanley[1], Mathew Williams[2], T. Luke Smallman[2], Edward Comyn-Platt[6], Peter Levy[7], Michel Ramonet[8], Paul I. Palmer[2].

[1]School of Chemistry, University of Bristol, Bristol, BS8 1TS, UK
[2]School of GeoSciences, University of Edinburgh, Edinburgh, EH9 3JN, UK
[3]School of Geographical Sciences, University of Bristol, Bristol, BS8 1SS, UK
[4]Hadley Centre, Met. Office, Exeter, EX1 3PB, UK
[5]Climate Science Centre, CSIRO Oceans and Atmosphere, Aspendale VIC 3195, Australia
[6]Centre for Ecology and Hydrology, Wallingford, OX10 8BB, UK
[7]Centre for Ecology and Hydrology (Edinburgh Research Station), Penicuik, E26 0QB, UK
[8]Laboratoire des Sciences du Climat et de l'Environnement, CEA-CNRS-UVSQ, Gif-sur-Yvette, 91198, France

*Correspondence to*: Emily White (emily.white@bristol.ac.uk), Matthew Rigby (matt.rigby@bristol.ac.uk)

**Abstract.** We present a method to derive atmospheric-observation-based estimates of carbon dioxide ($CO_2$) fluxes at the national scale, demonstrated using data from a network of surface tall tower sites across the UK and Ireland over the period 2013-2014. The inversion is carried out using simulations from a Lagrangian chemical transport model and an innovative hierarchical Bayesian Markov chain Monte Carlo (MCMC) framework, which addresses some of the traditional problems faced by inverse modelling studies, such as subjectivity in the specification of model and prior uncertainties. Biospheric fluxes related to gross primary productivity and terrestrial ecosystem respiration are solved separately in the inversion and then combined a posteriori to determine net primary productivity. Two different models, CARDAMOM and JULES, provide prior estimates for these fluxes. We carry out separate inversions to assess the impact of these different priors on the posterior flux estimates and evaluate the differences between the prior and posterior estimates in terms of missing model components. The Numerical Atmospheric dispersion Modelling Environment (NAME) is used to relate fluxes to the measurements taken across the regional network. Posterior $CO_2$ estimates from the two inversions agree within estimated uncertainties, despite large differences in the prior fluxes from the different models. With our method, averaging results from 2013 and 2014, we find a total annual net biospheric flux for the UK of $-8 \pm 79$ Tg $CO_2$ yr$^{-1}$ (CARDAMOM prior) and $-64 \pm 85$ Tg $CO_2$ yr$^{-1}$ (JULES prior), where -ve values represent an uptake of $CO_2$. These biospheric $CO_2$ estimates show that annual UK biospheric sources and sinks are roughly in balance. These annual mean estimates are consistently higher than the prior estimates, which show much more pronounced uptake in the summer months.



## 1 Introduction

There are significant uncertainties in the magnitude and spatiotemporal distribution of global carbon dioxide ($CO_2$) fluxes to and from the atmosphere, particularly those due to terrestrial biogenic systems (Le Quéré et al., 2018). Reliable methods for quantifying carbon budgets at policy relevant scales (i.e. national or sub-national) will be important if we are to accurately and transparently evaluate each country's progress towards achieving their Nationally Determined Contributions (NDCs) made following the Paris Agreement (UNFCCC, 2015).

Regional terrestrial carbon fluxes can be estimated using a range of observational, computational and inventory-based methods. These include "bottom-up" approaches such as the up-scaling of direct flux measurements made using eddy covariance or chamber systems (Baldocchi and Wilson, 2001) and models of atmosphere-biosphere $CO_2$ exchange. Flux measurements are important for understanding the small-scale processes responsible for carbon fluxes. However, they are relatively localised estimates (centimetres to kilometres), which are challenging to scale up to national levels. Biosphere models and land surface models (LSMs) can be used to estimate carbon fluxes using coupled representations of biogeophysical and biogeochemical processes, driven by observations of meteorology and ecosystem parameters (Potter, 1999; Clark et al., 2011; Bloom et al., 2016). Such models describe processes to varying degrees of complexity and are driven by observational data to varying degrees of detail; hence predictions of biogenic GHG fluxes can vary significantly between models (Todd-Brown et al., 2013; Atkin et al., 2015). Atmospheric inverse modelling is a "top-down" approach that provides an alternative to the bottom-up approaches described above. It has been used to indirectly estimate country-scale (e.g. Matross et al., 2006; Schuh et al., 2010; Meesters et al., 2012) and continental (e.g. Gerbig et al., 2003; Peters et al., 2010; Rivier et al., 2010) biospheric $CO_2$ budgets using atmospheric mole fraction observations, where the contribution of anthropogenic fluxes to the observations has been removed. In this approach, a model of atmospheric transport is used to relate spatiotemporally resolved surface fluxes of biospheric $CO_2$ to atmospheric measurements of $CO_2$ mole fractions. Biospheric fluxes derived from bottom-up approaches are often used as prior estimates in the inversion. Since atmospheric observations are sensitive to fluxes spanning tens to hundreds of kilometres (Gerbig et al., 2009), inverse methods are a valuable tool for examining national fluxes and evaluating estimates of surface exchange of $CO_2$ at larger spatial scales.

The United Kingdom (UK) government has set legally binding targets to curb greenhouse gas (GHG) emissions in an attempt to prevent dangerous levels of climate change. The Climate Change Act 2008 (The UK government, 2008) commits the UK to 80% cuts in GHG emissions, from 1990 levels, by 2050. To support this legislation, a continuous and automated measurement network has been established (Stanley et al., 2018; Stavert et al., 2018) with the goal of providing estimates of GHG emissions that are independent of the UK's bottom-up anthropogenic inventory that must be reported annually to the UK Parliament and submitted to the United Nations Framework Convention on Climate Change (UNFCCC). Previous studies have used data from the UK Deriving Emissions related to Climate Change (UK-DECC) network to infer emissions of



methane, nitrous oxide and HFC-134a from the UK (Manning et al., 2011; Ganesan et al., 2015; Say et al., 2016). These studies found varying levels of agreement with bottom-up inventory methods, where estimates of GHG emissions are made using reported statistics from various sectors (e.g. road transport, power generation, etc.). Here we use the DECC network and two additional sites from the Greenhouse gAs Uk and Global Emissions (GAUGE) programme (Palmer et al., 2018) to estimate biospheric fluxes of $CO_2$. Whilst anthropogenic emissions, which are the remit of the UK inventory, are not estimated in this

study, it represents the first step towards a framework for estimating the complete UK $CO_2$ budget.

Atmospheric inverse modelling of GHGs using Bayesian methods presents some known challenges. Robust uncertainty quantification in Bayesian frameworks can be difficult as they require that uncertainties in the prior flux estimate, and uncertainties in the model's ability to simulate the data, are well characterised. In practice, this is rarely the case and various

studies have investigated the use of data-driven uncertainty estimation (e.g. Michalak, 2004; Berchet et al., 2013; Ganesan et al., 2014; Kountouris et al., 2018b). Inversions are also known to suffer from "aggregation errors". One type of aggregation error arises from the way in which areas of the flux domain are grouped together to decrease the number of unknowns, because usually there are not sufficient data to solve for fluxes in each model grid cell (Kaminski et al., 2001). Furthermore, for reasons of mathematical and computational convenience, Gaussian probability density functions (PDFs) can be used to describe prior

knowledge (Miller et al., 2014). However, Gaussian assumptions can lead to unphysical solutions in the case of atmospheric GHG emissions or uptake processes, as they permit both positive and negative solutions.

$CO_2$ presents further complications over other GHGs, in that atmosphere-biosphere $CO_2$ exchange has a diurnal flux cycle that is significantly larger than the net flux, and has strong, spatially varying surface sources and sinks. Gerbig et al. (2003) was

one of the first to develop an analysis framework for regional scale $CO_2$ flux inversions. The study sets out the need to explicitly simulate the diurnal cycle of biospheric fluxes and highlights the importance of high spatial and temporal resolution data when addressing the unique problems of representation and aggregation errors caused by the highly varying nature of $CO_2$ fluxes in both space and time. Inverse modelling studies of $CO_2$ flux typically assume that anthropogenic fluxes are "fixed" in the inversion (e.g. Meesters et al., 2012; Kountouris et al., 2018a). This is based on the assumption that uncertainties in

anthropogenic fluxes are low compared to those of the biospheric fluxes. However, it has been suggested that this may not necessarily be the case (Peylin et al., 2011).

Here we outline a framework for evaluating the net biospheric $CO_2$ exchange (net ecosystem exchange, NEE) from a small to medium sized country using the high-resolution regional, Lagrangian transport model, the Numerical Atmospheric dispersion

Modelling Environment (NAME, Jones et al., 2006). To address many of the problems outlined above, we use an adapted form of a hierarchical Bayesian, trans-dimensional Markov chain Monte Carlo (MCMC) inversion (Rigby et al., 2011; Ganesan et al., 2014; Lunt et al., 2016). In the hierarchical Bayesian framework presented in Ganesan et al. (2014), "hyperparameters" that define the prior flux and model-data "mismatch" uncertainty PDFs are included in the inversion, which is solved using a





Metropolis-Hastings MCMC algorithm (e.g. Rigby et al., 2011). This hierarchical approach has been shown to lead to more
robust posterior uncertainty quantification in Bayesian frameworks where prior uncertainties are not well characterised
(Ganesan et al., 2014). Lunt et al. (2016) built on this method, developing a "trans-dimensional" framework that accounted for
the uncertainty in the definition of basis functions (the way in which flux grid cells are aggregated), and allowed this to
propagate through to the posterior estimate.

Gross primary productivity (GPP) and terrestrial ecosystem respiration (TER) estimates from the Joint UK Land Environment
Simulator (JULES) and CARbon DAta MOdel fraMework (CARDAMOM) are used as prior flux constraints. JULES is a
physically based, process driven model that estimates the energy, water and carbon fluxes at the land-atmosphere boundary
(Best et al., 2011; Clark et al., 2011). CARDAMOM, on the other hand, is a model-data fusion framework ingesting satellite
based remotely sensed estimates of state of terrestrial ecosystems to retrieve process parameters for the DALEC carbon balance
model (Bloom and Williams, 2015; Bloom et al., 2016; Smallman et al., 2017).

Below, we first describe our approach for modelling biospheric $CO_2$ fluxes, including several novel aspects compared to
previous work in this area. We then investigate the impact of using two different models that simulate biospheric fluxes
(JULES and CARDAMOM) within our proposed inverse framework and discuss the discrepancies between the prior and
posterior flux estimates.

## 2 Method

The main components of a regional atmospheric inverse modelling framework are the atmospheric $CO_2$ mole fraction data
itself, a model of atmospheric transport including a set of boundary conditions at the edge of the regional domain and some
initial information or "first guess" of regional $CO_2$ fluxes. These components are combined in an inversion set-up with a
mechanism for dealing with uncertainties in the inputs. To make the problem computationally manageable, the regional domain
is often decomposed into a number of basis functions, describing a spatial grouping of grid cells within which fluxes are scaled
up or down. The selection of these basis functions constitutes a further key element of the atmospheric inverse problem.

### 2.1 Site location and measurements

This study focuses on the years 2013 and 2014. During this period, atmospheric $CO_2$ mole fractions were continuously
measured at six sites across the UK and Republic of Ireland (see Table 1 for site information and Fig. 1 for the location of the
sites). Four of these sites originally formed the UK-DECC network and are described in Stanley et al. (2018), whilst two were
developed under the GAUGE programme and are described in Stavert et al. (2018). The site at Mace Head, Republic of Ireland,
is a coastal, 10 m above ground level (a.g.l.), station situated primarily to measure concentrations of background air arriving
at the site from the Atlantic Ocean. The Laboratoire des Sciences du Climat et de l'Environnement (LSCE) and the





Environmental Protection Agency (EPA) are responsible for making CO2 measurements at this site, with the support of the National University of Ireland Galway. Air is sampled from a 23 m.a.g.l. inlet (see Vardag et al., 2014 for a full site description). All of the UK sites are tall-tower stations (with inlets ranging from 42 to 248 m.a.g.l), designed to measure elevated greenhouse gas mole fractions as air is transported over the surface in the UK and Europe.

Continuous $CO_2$ measurements are made at all stations using cavity ring-down spectrometers (CRDS: Picarro G2301 or G2401). CRDS data are corrected for daily linear instrumental drift using standard gases and for instrumental non-linearity using calibration gases, spanning a range of above and below ambient mole fractions, on a monthly basis (Stanley et al., 2018). Calibration and standard gases are of natural composition and calibrated at the GasLab Max Planck Institute for Biogeochemistry, Jena, or the World Calibration Centre for $CO_2$ at Empa, linking them to the World Meteorological

Organisation (WMO) X2007 scale (Stanley et al., 2018; Stavert et al., 2018). At sites with multiple inlets, measurements are taken for the same length of time at each inlet, each hour. This means that measurements at each height at Bilsdale and Tacolneston (with 3 inlets) are taken continuously for roughly 20 minutes every hour, and at Heathfield and Ridge Hill (with 2 inlets) measurements are taken continuously for roughly 30 minutes at each inlet every hour. For the purposes of the inverse modelling carried out in this study, the continuous CRDS data are used from the highest measurement inlets and averaged to

a 2-hour time resolution. Further information about the instruments, measurement protocol and uncertainty estimates can be found in Stanley et al. (2018) and Stavert et al. (2018).

## 2.2 Atmospheric transport model

In this work we use a Lagrangian particle dispersion model (LPDM), NAME, which tracks thousands of particles back in time from observation locations. The model determines the locations where air masses interacted with the surface, and therefore

where surface $CO_2$ sources and sinks could contribute to a $CO_2$ concentration measurement. The model provides a gridded sensitivity of each mole fraction observation to the potential flux from each grid cell and this is often referred to as the "footprint" of a particular observation (for further details, see e.g. Manning et al., 2011).

At each two-hourly measurement time step, the model releases 20,000 particles, which are tracked back in time for 30 days,

so that by the end of this period the majority of particles will have left the model domain (Fig. S1). Since most $CO_2$ flux to the atmosphere occurs at the surface, we record the instances where the particles are in the lowest 40m of the atmosphere and assume that this represents the sensitivity of observed mole fractions to surface fluxes in the inversion domain. The domain used to calculate atmospheric transport covers most of Europe, the east coast of North and Central America and North Africa (-97.9° – 39.38° longitude and 10.729° – 79.057° latitude). The spatial resolution of the meteorological analysis dataset used

to drive the model, from the Met Office Unified Model (Cullen, 1993), was 0.233° by 0.352° (roughly 25 by 25 km over the UK).



In many previous inverse modelling studies using LPDMs (e.g. Manning et al., 2011; Thompson and Stohl, 2014; Steinkamp et al., 2017) the footprint is assumed to be equal to the integrated air history over the duration of the simulation (e.g. 30 days, as in Fig. 1). Based on the assumption that fluxes have not changed substantially during the 30-day period, the integrated footprint can be multiplied by the prior flux and summed over all the grid cells in the domain to create a time series of modelled mole fractions at each measurement site. However, many $CO_2$ inverse modelling studies using other LPDMs have disaggregated footprints back in time, capturing changes in surface sensitivity on timescales shorter than the duration of the simulation, thereby attempting to account for diurnal variation in $CO_2$ fluxes (Denning et al., 1996; Gerbig et al., 2003; Gourdji et al., 2010). Thus far, a disaggregation such as this has not been used in NAME simulations, so we describe our method here.

In our simulations, we determined the footprint for 2-hourly average periods back in time for the first 24 hours before the observation, and then replaced the first 24 hours of integrated sensitivities with these time-disaggregated footprints. Mole fractions were simulated by multiplying these footprints by biospheric flux estimates for the corresponding time, so that the variability in the source or sink of $CO_2$ was represented in the modelled observations. This is demonstrated in Eq. 1, which yields the modelled mole fraction, $y_t$, for one 2-hourly measurement time step, $t$, at one measurement site.

$$y_t = \sum_{i=0}^{12} \sum_{j=0}^{n} fp_{t-i,j} \times q_{t-i,j} + \sum_{j=0}^{n} fp_{remainder_j} \times q_{month_j} \qquad (1)$$

Here $i$ denotes the number of 2-hour periods back in time before the particle release at time $t$ and $j$ represents the grid cell where $n$ is the total number of grid cells; $fp_{t-i,j}$ is one grid cell of the two-dimensional time-disaggregated footprint for that time; $q_{t-i}$ is the one grid cell of the two-dimensional, two-hourly flux field corresponding to the time the particles were interacting with the surface; $fp_{remainder_j}$ is one grid cell of the remaining 29 day footprint and $q_{month_j}$ is one grid cell of the monthly average flux. We find that the diurnal cycle of $CO_2$ flux strongly impacts the mole fraction observations in the first 24 hours of transport before an observation is made. However, the use of time-integrated footprints, multiplied by average fluxes for the remainder of the period incurs only minor errors in our simulation. An investigation of the impact of going further back in time (e.g. 72 hours) revealed some small differences in daily maximum and minimum $CO_2$ concentration on some days in a forward model run (Fig. S2). Nevertheless, posterior UK net biosphere fluxes were in agreement, within the estimated posterior uncertainties, between an inversion using footprints disaggregated for 24 hours and one using footprints disaggregated for 72 hours (Table S1).

### 2.2.1 Data selection and model uncertainty

LPDMs are known to perform poorly under certain meteorological conditions. In particular, it is often assumed that model-data mismatch should be smallest during periods when the boundary layer is relatively well mixed. A common approach is to only include daytime data in the inversion (e.g. Meesters et al., 2012; Steinkamp et al., 2017; Kountouris et al., 2018a) or



separate morning and afternoon averages (e.g. Matross et al., 2006). To make use of as much high frequency measurement
information as possible, we use a filter based on two metrics to remove times of high atmospheric stability and/or stagnant
conditions. The first metric is based on calculating the ratio of the NAME footprint magnitude in the 25 grid boxes in the
immediate vicinity of the measurement station to the total for all of the grid boxes in the domain. A high ratio indicates times
when a significant fraction of air influencing the observation point originates from very local sources, which may not be

resolved by the model (Lunt et al., 2016). The second metric is based on the modelled lapse rate at each site, which is a measure
of atmospheric stability. A high lapse rate suggests very stable conditions, which would be conducive for significant local
influence. Thresholds for each of these criteria were chosen to preserve as much data as possible, whilst retaining only points
that the model was (somewhat subjectively) found to resolve well. In practice, the filter retained many more daytime than night
time points (see Fig. S3 for an analysis of the data removed in 2014) and inversion results were mostly similar to when only

daytime data were used, however differences were seen in some months when stagnant conditions occurred for several daytime
periods (Fig. S4).

Model uncertainty (or model-data mismatch) has a measurement uncertainty component and a component that takes into
account the ability of the model to represent real atmospheric conditions. The measurement uncertainty was assumed to be

equal to the standard deviation (st. dev.) of the measurements over the 2-hour period to give an estimate of measurement
repeatability and a measure of the sub-model-timescale variability in the observations. The 2-hourly measurement uncertainty
was then averaged over the month to ensure that measurements of high concentrations were not de-weighted, as they are more
likely to have greater variability and therefore a larger st. dev.. The measurement uncertainty is combined with a range of prior
values for model uncertainty (as this is a poorly constrained quantity) and together the model-measurement uncertainty is one

of the hyper-parameters solved in the inversion (further explained in Sect. 2.4.1).

### 2.2.2 Boundary conditions

The footprints from the LPDM only take into consideration the influence on the observations of sources intercepted within the
model domain. Therefore, an estimate of the mole fraction at the boundary must be made and incorporated into the simulated

mole fractions. To estimate spatial and temporal gradients in these boundary conditions we use the global Eulerian Model for
OZone And Related chemical Tracers (MOZART, Emmons et al., 2010). The model was run using GEOS-5 meteorology
(Rienecker et al., 2011) and global biospheric fluxes from the NASA-CASA biosphere model (Potter, 1999), global ocean
fluxes from Takahashi et al. (2009) and global anthropogenic fluxes from the Emission Database for Global Atmospheric
Research (EDGAR, EC-JRC/PBL, 2011). When particles leave the NAME model domain, we record the time and location of

the exit point. We then use MOZART to find the concentration of $CO_2$ at these locations to serve as prior boundary conditions.
The global MOZART initial mole fraction field for January 2014 was scaled before commencing the 2014 MOZART run to
match the surface South Pole value to the mean NOAA January 2014 flask value (Dlugokencky et al., 2018). This scaling



factor was also applied to any pre-January 2014 MOZART output to prevent any discontinuities in the boundary mole fraction fields. The mole fraction at each domain edge (N, E, S, W) is then scaled up or down during the inversion to account for

uncertainties in the MOZART boundary conditions (Lunt et al., 2016).

**2.3 Prior information**

In this work, we used model analyses to provide prior information about biospheric fluxes. Two models (CARDAMOM and JULES) were used to assess how much influence the choice of biospheric prior has on the outcome of the inversion. The

NAME model was used to simulate the contribution of anthropogenic and oceanic fluxes to the data, and this contribution was removed from the observations prior to the inversion. The fluxes used for this calculation are described below. The spatial and temporal resolution of the prior information is summarised in Table 3 and emissions from each source over the UK are shown in Figure 2.

In a synthetic data study in which biospheric $CO_2$ was inferred, Tolk et al., (2011) found that separately solving for positive fluxes (autotrophic and heterotrophic respiration combined, TER) and negative fluxes (GPP) in atmospheric inversions provided a better fit to the atmospheric mole fraction data than inversions that scaled NEE only. Equation 2 describes the relationship between these three variables:

$NEE = TER - GPP$        (2)

This separation has been applied in various studies demonstrating model set-ups with synthetic data, for example: geostatistical approaches (Göckede et al., 2010), ensemble Kalman filter methods (Zupanski et al., 2007; Lokupitiya et al., 2008) and Bayesian methods (Schuh et al., 2009). However, this separation is not routinely used in $CO_2$ inversions, as there are only a

limited number of real data studies where it has been implemented (e.g. Gerbig et al., 2003; Matross et al., 2006; Schuh et al., 2010; Meesters et al., 2012).

In this inversion, we separately solved for TER and GPP, and then combined them a posteriori to determine NEE. Similarly to the studies cited above, we find closer agreement with the data than if NEE were scaled directly. Furthermore, we note that, if

only one factor is used to scale both TER and GPP, it is impossible for the inversion to respond to a prior that has, for example, too strong a sink but a source of the correct magnitude. To demonstrate this, we have carried out a synthetic test (Fig. S5) in which we have investigated the ability of our inversion system to resolve a "true" flux, created using the CARDAMOM prior fluxes but with a GPP that has been halved, in an inversion that used the full CARDAMOM fluxes as a prior. Figure S5(a) shows that the seasonal cycle of posterior fluxes for the inversion where GPP and TER are separated is able to change phase

and manages to approach the "true" flux. In contrast, the seasonal cycle of posterior fluxes for the inversion where NEE is



scaled can only move towards the "true" flux by shallowing the amplitude without changing the phase, because reducing the sink means the source must also be reduced. This can also be seen in the posterior diurnal cycles of GPP, TER and NEE, which are shown as an average for June in Fig. S5(b) and Fig. S5(c). The inversion that separates the two sources is able to correctly estimate the diurnal cycle of the "true" GPP flux, which allows it to find a better estimate for the diurnal cycle of NEE than

the case where NEE is scaled. Whilst this is an extreme case, a comparison of the CARDAMOM and JULES estimates demonstrates large relative differences in TER and GPP, which would be "hard wired" if only NEE were scaled (Figure S5(b)).

Given the results of our simple synthetic test, separating GPP and TER in the inversion appears to be an important improvement on scaling NEE directly and it is what we have implemented here. However, in addition to the main inversions presented in

this paper, where GPP and TER are separated, we have carried out two further inversions for JULES and CARDAMOM where only NEE is scaled. The results of these additional inversions are discussed in Sect. 4.1.

### 2.3.1 CARDAMOM biospheric fluxes

The CARbon DAta MOdel fraMework (CARDAMOM) uses a model-data fusion approach to retrieve location specific ensembles of parameters for the DALEC model (Bloom et al., 2016). CARDAMOM uses a Bayesian approach within a

Metropolis-Hastings MCMC algorithm to compare model states and flux estimates against observational information to determine the likelihood of potential parameter sets guiding the parameterisation processes. DALEC simulates the ecosystem carbon balance, including uptake of $CO_2$ via photosynthesis, $CO_2$ loss via respiration, mortality and decomposition processes, and carbon flows between ecosystem pools (non-structural carbohydrates, foliage, fine roots, wood, fine litter, coarse woody debris and soil organic matter). GPP or photosynthesis is estimated using the aggregated canopy model (ACM; Williams et

al., 1997) while autotrophic respiration is estimated as a fixed fraction of GPP. Mortality and decomposition processes follow first order kinetic equations (i.e. a daily fractional loss of the C stock in question). The mortality and decomposition parameters are modified based on an exponential temperature sensitivity parameter. Note that the current version of DALEC does not include a representation of the water cycle, rather water stress is parameterised through a sensitivity to high vapour pressure deficit. Comprehensive descriptions of CARDAMOM can be found in Bloom et al. (2016) and Bloom and Williams (2015)

and DALEC in Smallman et al. (2017).

The CARDAMOM analysis to generate the carbon flux priors was conducted at a weekly time step and 25 km × 25 km spatial resolution. The weekly time step information was downscaled to 2-hourly intervals, assuming that each day repeated throughout each week. Downscaling of GPP fluxes was assumed to be distributed through the daylight period based on

intensity of incoming shortwave radiation. Respiration fluxes were downscaled assuming exponential temperature sensitivity (code for downscaling is available from the authors on request).



Observational information used in the current analysis are satellite-based remotely sensed time series of Leaf Area Index (LAI) (MODIS; MOD15A2 LAI-8 day version 5, http://lpdacc.usgs.gov/) a prior estimate of above ground biomass (Thurner et al.,
2014) and a prior estimate of soil organic matter (Hiederer and Köchy, 2012). Meteorological drivers were taken from ERA-Interim reanalysis. Ecosystem disturbance due to forest clearances were imposed using Global Forest Watch information (Hansen et al., 2013).

### 2.3.2 JULES biospheric fluxes

The Joint UK Land Environment Simulator (JULES) is a process driven Land Surface Model (LSM) that estimates the energy,
water and carbon fluxes at the land-atmosphere boundary (Best et al., 2011; Clark et al., 2011). We used JULES version 4.6 driven with the WATCH Forcing Data methodology applied to Era-Interim reanalysis data (WFDEI) meteorology (Weedon et al., 2014) which were interpolated to a $0.25° \times 0.25°$ (Schellekens et al., 2017). We prescribed the land cover for 9 surface types and the vegetation phenology for 5 plant functional types (PFTs) using MODIS monthly LAI climatology and fixed MODIS land cover and canopy height data (Berry, et al., 2009). The soil thermal and hydrology physics are described using
the JULES implementation of the Brooks and Corey formulation (Marthews et al., 2015) with the soil properties sourced from the Harmonised World Soil Database (FAO/IIASA/ISRIC/ISS-CAS/JRC, 2009). Soil carbon was calculated as the equilibrium balance between litter-fall and soil respiration for the period 1990-2000 using the formulation of (Mariscal, 2015). The full JULES configuration and science options are available for download from the Met Office science repository (https://code.metoffice.gov.uk/trac/roses-u/browser/a/x/0/9/1/trunk?rev=75249).

### 2.3.2 Anthropogenic fluxes

Estimates of fluxes due to anthropogenic activity within the UK were obtained from the National Atmospheric Emissions Inventory (NAEI, http://naei.beis.gov.uk). The NAEI provides a yearly estimate of emissions, which we have disaggregated into a 2-hourly product, based on temporal patterns in activity data, varying on diurnal, weekly and seasonal scales. The inventory emissions were disaggregated according to the UNECE/CORINAIR Selected Nomenclature for sources of Air
Pollution (SNAP) sectors (UNECE/EMEP, 2001). Figure 2(d) shows the seasonal and diurnal cycle for this inventory, summed over the UK, for 2014. Outside the UK, prior anthropogenic emissions come from the Emission Database for Global Atmospheric Research (EDGAR) v4.2 FT2010 inventory data for 2010 (EC-JRC/PBL, 2011). This is a fixed 2D map that is used throughout the inversion period.

### 2.3.3 Ocean fluxes

Prior ocean flux estimates are from Takahashi et al. (2009). They are based on a climatology of surface ocean pCO$_2$ constructed using measurements taken between 1970 and 2008. The monthly UK coastal ocean flux (defined simply as the grid cells closest to the UK, up to a maximum distance of 500km) from this product is plotted in Fig. 2(e).



### 2.4 Inverse method

### 2.4.1 Hierarchical trans-dimensional Bayesian inversion

Like many atmospheric inverse methods, our framework is based on traditional Bayesian statistics, given by Eq. 3:

$$\rho(\boldsymbol{x}|\boldsymbol{y}) = \frac{\rho(\boldsymbol{y}|\boldsymbol{x})\rho(\boldsymbol{x})}{\rho(\boldsymbol{y})} \qquad (3)$$

where $\boldsymbol{y}$ is a vector containing the observations and $\boldsymbol{x}$ is a vector of the parameters to be estimated (such as the flux and
boundary condition scaling). The traditional Bayesian approach requires that decisions about the form of the prior PDF, $\rho(\boldsymbol{x})$,
and likelihood function, $\rho(\boldsymbol{y}|\boldsymbol{x})$, are made a priori. These pre-defined decisions have the potential to strongly influence the
form of the posterior PDF in an inversion (Ganesan et al., 2014). Instead, we introduce a second "level" to the traditional Bayes
equation, to account for the fact that initial parameter uncertainty estimates are themselves uncertain. This is known as a
"hierarchical" Bayes framework where additional parameters, known as hyper-parameters, are used to describe the
uncertainties in the prior and the model.

Alongside the additional hyper-parameters $\boldsymbol{\theta}$, we also introduce an additional term, $k$, that describes the size of the inversion
grid, following the trans-dimensional inversion approach described in Lunt et al. (2016). In this approach, the number of basis
functions to be solved is not fixed a priori and hence $\boldsymbol{x}$ has an unknown length. The number of unknowns is itself a parameter
to be solved for in the inversion, with the uncertainty in this term propagating through to the posterior parameter estimates,
more fully accounting for the uncertainties that are only tacitly implied within a traditional Bayesian approach. The full trans-
dimensional hierarchical Bayesian equation that is solved in our inversion thus becomes:

$$\rho(\boldsymbol{x},\boldsymbol{\theta},k|\boldsymbol{y}) \propto \rho(\boldsymbol{y}|\boldsymbol{x},\boldsymbol{\theta},k)\rho(\boldsymbol{x}|\boldsymbol{\theta},k)\rho(k)\rho(\boldsymbol{\theta}) \qquad (4)$$


where θ is a set of hyper-parameters describing the uncertainty on $\boldsymbol{x}$ ($\sigma_x$), the model-measurement error ($\sigma_y$), and the
correlation timescale in the model-measurement covariance matrix ($\tau$). These hyper-parameters are summarised in Table 2
along with the prior PDFs used to describe them in this inversion set-up.

In this study, we have adapted the trans-dimensional method to keep a fixed set of regional basis functions (described in Sect.
2.4.3) but allow the inversions to have a variable *time* rather than *space* dimension. We perform our inversion calculations
over one month at a time, but with the trans-dimensional case in *time* we find multiple scaling factors for each fixed region
over the course of the inversion, down to a minimum daily resolution. Therefore, in this case $k$ is more specifically the



unknown number of time periods resolved in the inversion, which is important for the highly variable temporal nature of $CO_2$
fluxes.

In general, there is no analytical solution to our hierarchical Bayesian equation, so we approximate the posterior solution using
a reversible jump Metropolis-Hastings MCMC algorithm (Metropolis et al., 1953; Green, 1995; Tarantola, 2005; Lunt et al.,
2016). The algorithm explores the possible values for each parameter by making a new proposal for a parameter value at each
step of a "chain" of possible values. Proposals are accepted or rejected based on the a comparison between the "current" and
"proposed" state's fit to the data (likelihood ratio), deviation from the prior PDF (prior ratio), and a term governing the
probability of generating the proposed state versus the reverse proposal (proposal ratio). More favourable parameter values or
model states are always accepted; however, less favourable parameter values or model states can be randomly accepted in
order to fully explore the full posterior PDF. The algorithm had a burn-in period of $5 \times 10^4$ iterations and was run for $2 \times 10^5$
iterations to appropriately explore the posterior distribution. At the end of the algorithm a chain of all accepted parameter
values is stored (if a proposal is rejected the chain will spend longer at the previously accepted value). A histogram of this
chain describes a posterior PDF for each parameter so that statistics such as the mean, median and standard deviation can be
calculated. The trace of each chain was examined qualitatively to ensure that the algorithm had been run for a sufficient number
of iterations to converge on a result.

**2.4.2 Basis functions**

Our domain is split into five spatial regions separating West-Central Europe from North-East, South-East, South-West and
North-West regions, shown in Fig. S1. Within the West-Central Europe area (the hashed region in Fig. S1), a map of the
fraction of different plant functional types (PFTs) in each grid cell has been used to further break down the region. This is the
same PFT map used in the JULES biospheric simulation (see Sect. 2.3.2). A scaling factor is solved in the inversion, scaling
GPP and TER within maps of five or six PFTs: broadleaf tree, needleleaf tree, C3 grasses, C4 grasses, shrubland and, in the
case of TER, bare soil.

**2.4.3 Definition of Jacobian matrix**

Footprints from NAME, prior fluxes, boundary conditions and basis functions are all combined into a matrix of partial
derivitives, alternatively described as a "Jacobian" or "sensitivity" matrix, that describes the change in mole fraction with
respect to a change in each of the input parameters. This is the "model" in the inversion set-up, denoted **H** in the description
of the linear forward model (Eq. 7), where $\boldsymbol{\varepsilon}$ is the mismatch between modelled "observations" and what has actually been
measured in the atmosphere. **H** has dimensions $m$ (number of data points) by $n$ (number of basis functions).

$$\boldsymbol{y} = \mathbf{H}\boldsymbol{x} + \boldsymbol{\varepsilon} \qquad (7)$$




To create this model, we multiplied the footprints by the prior GPP and TER fluxes separately, then multiplied these by the fractional map of basis functions (described in Sect. 2.4.2) and summed over the area covered by each basis function. The boundary conditions were broken down by four further basis functions for each edge of the domain as explained in Sect. 2.2.2. Multiplying the sensitivity matrix by a vector of ones yields the prior modelled time-series at a site. Therefore, during our

inversion, we are updating this vector of ones as a scaling factor, to scale up or down emissions for each PFT and biospheric component to better agree with the data. Whilst in theory we have posterior information about the gross GPP and TER biospheric components separately, we combine this into a net ecosystem exchange (NEE) flux estimate, as we believe this to be more robust (Tolk et al., 2011). Therefore, throughout this paper we discuss posterior NEE estimates, however the results of the separate sources can be found in the supplement in Fig. S5-S7.

## 3 Results

We have tested our $CO_2$ inversion set-up using output from two different models of biospheric flux as a prior constraint and find estimates for UK biospheric $CO_2$ flux from these two inversions. We first describe differences between the output from the two prior models, then present the UK flux estimates found with this method, along with the spatial distribution of posterior fluxes.

### 3.1 Differences between CARDAMOM and JULES

The $CO_2$ fluxes from CARDAMOM and JULES differ both temporally and spatially. Figure 2 (a-c) shows UK fluxes of GPP, TER and NEE from the two models. Most notable differences are seen in TER where JULES has a large diurnal range whereas CARDAMOM has a small diurnal range. Averaged to monthly resolution, the fluxes are relatively similar although CARDAMOM has a higher TER flux from July to October. Diurnal ranges for GPP are more similar in magnitude, however

JULES exhibits a stronger sink in spring with maximum uptake in June. CARDAMOM has maximum uptake in July and exhibits a stronger sink in autumn. Combining these two fluxes, we can see that the profile of NEE for both models is quite different. The daily maximum source from JULES remains relatively constant throughout the year, whereas the daily maximum source in CARDAMOM follows a similar seasonal cycle to the daily maximum sink (albeit with a smaller magnitude). Monthly net fluxes are similar between both models for much of the year although JULES has stronger uptake between March and June.


In order to understand some of these seasonal differences we can compare the processes taking place in each model. The CARDAMOM system explicitly simulates the soil and litter stocks, growth and turnover processes. LAI is retrieved from the DALEC model (which was calibrated using MODIS LAI estimates at the correct time and location of the analysis, explained in Sect. 2.3.1) and updated at each weekly time step. In the JULES system, soil and litter carbon stocks are fixed values for

each grid cell, calibrated from 1990-2000, and it uses a fixed climatology of LAI and canopy height. Therefore, variability in TER and GPP fluxes from JULES are governed by meteorology, primarily temperature but also significant signals from



photosynthetically active radiation and precipitation via the soil moisture. This gives CARDAMOM interannual variability in LAI and soil carbon stocks, whereas these parameters remain constant in JULES year to year. However, meteorology drives the JULES model at a 2-hourly timestep as opposed to a weekly time-step in CARDAMOM. Therefore, in the 2-hourly
CARDAMOM product used here, the diurnal range is not explicitly simulated and is downscaled from a weekly resolution. This downscaling is done based on light and temperature curves as explained in Sect. 2.3.1. In both models, the autotrophic respiration is a fixed fraction of the GPP, roughly ranging from 0.1 to 0.5. Therefore, there are some large differences between the model processes, particularly in how the respiration fluxes are simulated. This could be leading to too small a diurnal range in CARDAMOM TER and too large a diurnal range in JULES TER.


Figures 3 and 4 show spatial maps of GPP, TER and NEE from both models averaged over winter (December, January, February) and summer (June, July, August) months. The pattern of TER is similar for both models, however JULES always has a stronger source over Northern Ireland and CARDAMOM has a stronger source in east England. In winter there are only small spatial variations in CARDAMOM GPP fluxes, whereas JULES has its largest uptake in south-west England and Wales.
In summer, the models are roughly in agreement in the size of the sink in Wales and the majority of England, however JULES has a stronger sink in Scotland and Northern Ireland and CARDAMOM has a stronger sink in central and south-east England. The differences between the models in GPP and TER lead to fairly different winter NEE flux maps. CARDAMOM is a net source everywhere in winter, with areas of strongest net source in southern Scotland, east and central England. JULES is a small net winter sink in Northern Ireland, Wales, and south and central England. Summer NEE fluxes are similar between the
models, although JULES has a stronger net sink in Scotland and Northern Ireland.

## 3.2 Posterior net UK biospheric CO₂ flux 2013-2014

We have derived estimates for annual NEE from the UK using $CO_2$ flux output from the two different models of biospheric flux as prior information (Fig. 5 – blue and orange bars for CARDAMOM and JULES respectively): $-13\pm^{90}_{87}$ Tg $CO_2$ yr$^{-1}$ (CARDAMOM prior) and $-76\pm^{91}_{90}$ Tg $CO_2$ yr$^{-1}$ (JULES prior) in 2013 and $-2\pm^{70}_{68}$ Tg $CO_2$ yr$^{-1}$ (CARDAMOM) and $-51\pm^{80}_{78}$
Tg $CO_2$ yr$^{-1}$ (JULES) in 2014. These annual net flux estimates from both models agree within the estimated uncertainties and mean values are higher than their respective priors in both cases. The uncertainties straddle the zero net flux line implying that the UK is roughly in balance between sources and sinks of biospheric $CO_2$. However, according to the inversion using JULES, a net biospheric source is less likely than in the inversion using CARDAMOM. When added to the anthropogenic and ocean fluxes that remained fixed during the inversion we produce the following estimates for annual total net $CO_2$ release from
the UK (Fig. 5 – green and yellow bars for CARDAMOM and JULES respectively): $448 \pm^{90}_{87}$ Tg $CO_2$ yr$^{-1}$ (CARDAMOM prior) and $386 \pm^{91}_{90}$ Tg $CO_2$ yr$^{-1}$ (JULES prior) in 2013 and $418 \pm^{70}_{68}$ Tg $CO_2$ yr$^{-1}$ (CARDAMOM prior) and $369 \pm^{80}_{78}$ Tg $CO_2$ yr$^{-1}$ (JULES prior) in 2014. While we are assuming that anthropogenic and ocean fluxes are perfectly known, the uncertainties on these fluxes are comparatively small (Peylin et al., 2011). When the anthropogenic source was varied by $\pm 10\%$, a conservatively large estimate of these uncertainties, we find posterior biospheric flux estimates using the CARDAMOM prior





that still suggest a balanced biosphere, and posterior flux estimates using the JULES prior that suggest a small net sink at the lowest end of the possibilities explored here (see Fig. S11). All mean annual posterior estimates, regardless of the anthropogenic source used, suggest the prior net biospheric flux is underestimated, i.e. posterior biospheric uptake of $CO_2$ is smaller than predicted by the models. However, this is less statistically significant with the 2013 inversion using the CARDAMOM prior.


The monthly posterior UK estimates using both models (Fig. 5) mostly agree well with each other within the uncertainties, however they are both notably different from the prior estimates, especially in 2014. The posterior total UK flux estimate, achieved by adding the posterior NEE fluxes to anthropogenic and coastal ocean fluxes, shows that, according to the CARDAMOM inversion, the UK may not be a net sink of $CO_2$ at any time of year in 2013 and 2014. However, the JULES

inversion suggests the UK is a net sink of $CO_2$ in June of both years.

Posterior seasonal cycle amplitudes are generally smaller than the prior amplitudes, except in the CARDAMOM inversion in 2014. Table 5 gives the posterior maximum and minimum values of NEE, leading to seasonal cycle amplitudes of 469 Tg $CO_2$ $yr^{-1}$ and 578 Tg $CO_2$ $yr^{-1}$ for 2013 and 633 Tg $CO_2$ $yr^{-1}$ and 737 Tg $CO_2$ $yr^{-1}$ for 2014, for the CARDAMOM and JULES

inversions respectively. These values are 90% and 76% of the prior amplitudes in 2013 and 123% and 85% of the prior amplitudes in 2014.

The largest differences between the prior and posterior are seen in spring and summer for both models. Posterior UK NEE estimates from the CARDAMOM inversion are in agreement with the prior for 11 months: during the first half of 2013, in the

majority of winter months (December, January, February) and in June 2014. When the CARDAMOM inversion posterior UK NEE estimates are not in agreement with the prior, they are usually larger, with a maximum difference in 2013 of $235 \pm^{92}_{91}$ Tg $CO_2$ $yr^{-1}$ in August and a maximum difference in 2014 of $551 \pm^{80}_{84}$ Tg $CO_2$ $yr^{-1}$ in July, although in spring (March, April, May) 2014 they tend to be smaller than the prior, with a maximum difference of $-194 \pm^{64}_{60}$ Tg $CO_2$ $yr^{-1}$ in April. Posterior UK NEE from the JULES inversion agrees with the prior for nine months during the two-year period, the majority of which are between

November and February. Otherwise, the posterior estimate from the JULES inversion is larger than the prior with a maximum difference in 2013 of $318\pm^{70}_{71}$ Tg $CO_2$ $yr^{-1}$ in April and a maximum difference in 2014 of $407 \pm^{76}_{72}$ Tg $CO_2$ $yr^{-1}$ in July.

Looking at the spring and summer differences more closely, we find that the JULES model has a systematically lower net spring flux than the posterior, and the CARDAMOM model is either in agreement with or higher than the posterior estimate

of the net spring flux. Generally the models are underestimating the net summer flux (to the greatest extent in 2014) although the summer estimate from the JULES inversion in 2013 is not statistically different from the prior. The average spring difference between the posterior and the prior for the CARDAMOM inversion is $-2 \pm^{89}_{88}$ Tg $CO_2$ $yr^{-1}$ in 2013 and $-133 \pm^{67}_{63}$ Tg $CO_2$ $yr^{-1}$ in 2014, whereas for the JULES inversion it is $219 \pm 87$ Tg $CO_2$ $yr^{-1}$ in 2013 and $164 \pm^{67}_{65}$ Tg $CO_2$ $yr^{-1}$ in 2014.



The average summer difference for the CARDAMOM inversion is $135 \pm_{108}^{111}$ Tg $CO_2$ yr$^{-1}$ in 2013 and $263 \pm_{83}^{82}$ Tg $CO_2$ yr$^{-1}$ in
2014, whereas for the JULES inversion it is $94 \pm_{107}^{104}$ Tg $CO_2$ yr$^{-1}$ in 2013 and $312 \pm 85$ Tg $CO_2$ yr$^{-1}$ in 2014. The prior sink
in June as estimated by the JULES model is nearly twice that of CARDAMOM and posterior estimates tend to agree with the
CARDAMOM prior in this month.

Figure S8(c) shows the daily minimum and maximum in the posterior net biospheric estimates for 2014. It is worth bearing in
mind at this point that while the temporal resolution of the inversion is flexible, it can go down to a minimum resolution of
one day (as explained in Sect. 2.4.1). Therefore, the diurnal profile of TER and GPP for each model is imposed, however it
can be scaled up or down from day to day. For both inversions, the posterior NEE flux has a similar profile. Compared to Fig.
2(c) the inversion tends to a seasonal cycle in daily maximum uptake that resembles that of the JULES model prior, with a
turning point in maximum uptake occurring abruptly between June and July, a steep gradient in spring and a shallow gradient
in autumn. On the other hand, the seasonal cycle in daily maximum source resembles that of the CARDAMOM model prior,
which has a stronger seasonal variation compared to that of the JULES model prior, albeit with a larger amplitude. This would
suggest that the underestimation in net spring flux seen in the JULES prior is generally due to the model underestimating the
spring source, rather than overestimating the spring sink. It also suggests that the overestimation in net summer flux in the
CARDAMOM prior is possibly a combination of the model overestimating the summer sink and underestimating the summer
source. The overestimation in the net summer flux in JULES is more likely to be due to an underestimation of the summer
source. However, as diurnal fluxes vary on a scale nearly an order of magnitude larger than that of the monthly fluxes, it is
clear that any relatively small changes in the maximum source or sink will have a relatively large effect on the daily net flux.
Therefore, the monthly net flux is the more robust result here and we are not able to confidently draw conclusions from the
sub-monthly results.

## 3.3 Posterior spatial distribution of biospheric fluxes

Figure 6 shows mean posterior net biospheric fluxes (NEE) for winter 2013 and summer 2014 from both the CARDAMOM
and JULES inversions. In winter 2013, posterior NEE fluxes from the CARDAMOM inversion are fairly heterogeneous and
are largest over south-west Scotland and east and central England. This posterior spatial distribution is roughly similar to the
prior. From the inversion using JULES prior fluxes, the posterior net biospheric flux is much smoother than it is for the
inversion using CARDAMOM. It is largest in north-west England and almost zero in east England. The whole of south/central
England, Wales, and Northern Ireland have increased posterior winter fluxes compared to the prior, turning these areas from
a net sink in the prior to a net source in the posterior.

In summer 2014, NEE fluxes from the two inversions display many similarities, with areas of net source in east, central
(extending further south in the JULES inversion) and north-west England and areas of net sink elsewhere. However, the net
sink in JULES is larger than CARDAMOM in Scotland, south Wales, Northern Ireland and south-west England. This differs





from the prior flux maps, which have only very small areas of small net uptake in central England in CARDAMOM and in east England in JULES. Both the CARDAMOM and JULES posterior fluxes generally display reduced uptake compared to the prior, except in north Wales.

**3.4 Model-data comparison**

Simulated mole fractions are greatly improved after the inversion, with $R^2$ values increasing by a minimum of 0.29 and up to 0.51 (to give values ranging between 0.54 and 0.74) and root mean square error (RMSE) decreasing by at least 0.29 ppm and up to 2.51 ppm (to give values ranging between 1.34 ppm and 2.56 ppm). Table S2 shows all statistics for the prior and posterior mole fractions compared to the observations of atmospheric $CO_2$ concentration. In terms of $R^2$, the best fit to the data is observed at Heathfield in the CARDAMOM inversion and Angus in the JULES inversion. In terms of RMSE, the best fit to the data is observed at Angus in the CARDAMOM inversion and Mace Head in the JULES inversion. The smallest posterior mean bias is observed at Angus in the CARDAMOM inversion and Ridge Hill in the JULES inversion. Figures S9 and S10 show the residual mole fractions in 2014. The Figures show that residuals are somewhat larger during the summer than the winter.

**4 Discussion**

**4.1 Inversion performance**

Solving for both TER and GPP separately allows the JULES-prior and CARDAMOM-prior inversions to converge to a similar posterior solution. Using two very different prior NEE flux estimates, we produce two similar posterior NEE flux estimates that have a similar seasonal amplitude, and agree on the majority of monthly and all annual fluxes within the estimated uncertainties. This indicates that our results are driven by the data rather than determined by the prior. However, when we carry out the same inversion but scale NEE (Fig. S12) we find the two posterior flux estimates do not converge on a common result. The posterior seasonal cycles remain relatively unchanged compared to the prior and annual net biospheric flux estimates tend to be similar to, or larger than, the prior. These annual net biospheric flux estimates are therefore 3 – 39 times smaller than the inversion that separates GPP and TER, meaning the posterior estimates from the two types of inversions do not overlap, even within estimated uncertainties. Evaluating the statistics of how well the NEE inversions fit the data (Table S3), we find they do not perform as well as the separate GPP and TER inversion. However, this is to be expected to some degree, because separating the two sources gives the inversion more degrees of freedom to fit the data.

As recommended by Tolk et al. (2011), we are only hoping to achieve an improved estimate for the net fluxes here rather than the gross GPP and TER fluxes themselves. The posterior gross fluxes are included in the supplement (Fig. S6-S8) but due to the correlation between the spatial and temporal distribution of GPP and TER they have not been presented in the main text. This can be seen in summer and winter flux maps (Fig. S5 and S6) and in the posterior annual flux estimates in Fig. S8(d), in



particular where JULES TER and GPP show similarly large differences from the prior. This could also be a result of the imposed diurnal cycle, as it would appear the posterior TER flux in the JULES inversion is tending to a higher daily minimum,
matching that of the CARDAMOM prior, and may ultimately be trying to move towards a smaller diurnal variation in TER. However, because the whole diurnal cycle must be scaled, the daily maximum TER must also increase and may mean the GPP must increase, causing increased uptake, to compensate for the increased source from TER. Allowing flexibility on sub-daily timescales may lead to similar estimates of GPP and TER between the two inversions with different priors. However, questions remain over whether there is enough temporal information for this to be the case.


The fact that common monthly and annual posterior net biospheric flux estimates are reached when the prior biospheric fluxes are spatially and temporally different would suggest that the choice of prior is not necessarily a major factor in guiding the inversion result for our network, when GPP and TER are scaled separately. In this respect, it is also particularly encouraging that the seasonal cycles in the posterior diurnal range are similar for both inversions (Fig. S8(c)).

**4.3 Differences between prior and posterior NEE estimates**

The posterior seasonal cycle in both inversions differs significantly from the prior. This implies that the biospheric models used to obtain prior GPP and TER fluxes are either over- or under-estimating the strength of some processes, or they are omitting some processes altogether. The largest differences between the posterior solution and the prior model output are seen in spring and summer. In Sect. 3.2 we have shown that spring differences arise from an underestimation of the net spring flux
in the JULES model and a correct/overestimation of the net spring flux in CARDAMOM. However, in summer (particularly in 2014), the posterior net UK fluxes are higher than both priors in July and August.

One process that occurs during the months July and August is crop harvest. This is not taken into account in either of the models of the biosphere used in this work, thereby providing a possible explanation for the differences between the posterior
and prior in these months. Harvest typically occurs between July and September and arable agricultural land covered 26% of the UK in 2013 and 2014 (DEFRA, 2014, 2015), so there is potential for unaccounted activity in this area to cause large changes to net $CO_2$ fluxes. The areas of net source in summer (shown in Fig. 6) do also coincide with areas of large-scale agriculture (e.g. east and central England). Crop harvest potentially changes the biosphere in the following ways: firstly, crops mature en masse, leading to an abrupt loss of productivity. Secondly, during harvest there is an abrupt removal of biomass and
input of harvest residues on the field. This increases litter input that is readily available for decomposition, increasing heterotrophic respiration. Thirdly, when the field is ploughed the soil is disturbed, which will again increase heterotrophic respiration. Finally, when the crop is no longer covering the soil surface this layer can become drier and the energy balance is altered. In Smallman et al. (2014), the reduction in atmospheric $CO_2$ concentration due to crop uptake is reported for 2006 to 2008 and an abrupt increase in atmospheric $CO_2$ can be seen between June (peak source) and August, where $CO_2$ uptake from
crops is halted as a result of harvest. This may explain the abrupt shift from net sink to net zero / net source observed between

July and August in CARDAMOM in 2013 and June and July in both models in 2014. The earlier time in 2014 does coincide with a year of early harvest (DEFRA, 2015) although this may well be fortuitous. Later in the summer, there may be some plant regrowth in ploughed fields leading to increased GPP. This would be consistent with the shallower gradient observed in net biospheric fluxes between September and October 2013 in the CARDAMOM posterior estimate, between August and
September 2014 in the JULES posterior estimate and the decrease in net flux observed between July and September 2014 in the CARDAMOM posterior estimate.

If agricultural activity is the source of the July, August, September difference between prior and posterior UK NEE estimates, then it could amount to emissions of 4 – 10 % of currently reported annual anthropogenic emissions in 2013 and 17 – 19 % in
2014. However, other explanations for this difference could be large uncertainties in the seasonal disaggregation of anthropogenic fluxes, uncertainties in the transport model, or a combination of over-/under-estimation of other biospheric processes.

### 4.2 Implications for UK CO$_2$ emissions estimates

The results of UK biospheric CO$_2$ fluxes using our set-up suggest the UK biosphere is roughly in balance, whereas prior
estimates from models of the biosphere estimate a net sink. Even when we assume an uncertainty on our anthropogenic fluxes of 10% (a conservative estimate), inversions using both models still give mean posterior estimates that are larger than their respective priors (see Fig. S11). Therefore, when using models of the biosphere to contribute to inventory estimates of CO$_2$ emissions, care must be taken to attribute sufficient uncertainties to model estimates, otherwise the amount of CO$_2$ taken up by the biosphere on an annual basis may be overestimated. Methods, such as the one described in this paper, could provide an
important constraint on the UK's biospheric CO$_2$ fluxes as carbon sequestration processes, such as reforestation, and other land use change activities are increasingly used as policy solutions to contribute to carbon targets.

### 5 Conclusion

We have developed a framework for estimating net biospheric CO$_2$ fluxes in the UK that takes advantage of recent innovation in atmospheric inverse modelling and a relatively dense regional network of measurement sites. Two inversions are carried
out using prior flux estimates from two different models of the biosphere, CARDAMOM and JULES. Fluxes of GPP and TER are scaled separately in the inversions. Despite significant differences in prior biospheric fluxes, we find consistent monthly and annual posterior flux estimates, suggesting that the choice of model to provide biospheric CO$_2$ flux priors in the inversion is not a major factor in guiding the inversion result with our framework and network.



Similarly to Tolk et al. (2011), we find that the NEE is more robustly derived if GPP and TER are solved separately, and then combined a posteriori. Our results suggest that inversions that scale only NEE could be underestimating net $CO_2$ fluxes, as we find posterior estimates 3 – 39 times smaller than those obtained using an inversion where GPP and TER are separated.

We find that the UK biosphere is roughly in balance, with annual net fluxes (averaged over the study period) of $-8 \pm 79$ Tg
$CO_2$ yr$^{-1}$ and $-64 \pm 85$ Tg $CO_2$ yr$^{-1}$ according to the CARDAMOM and JULES inversions respectively. These mean annual fluxes are systematically higher than their respective priors, implying that net biospheric fluxes are underestimated in the models of the biosphere used in this study. The posterior seasonal cycles from both inversions differ significantly from the prior seasonal cycles and generally have a reduced amplitude of 90% and 76% of the prior amplitude in 2013 according to the CARDAMOM and JULES inversions respectively, and 85% of the prior amplitude in 2014 according to the JULES inversion,
however the posterior seasonal cycle amplitude from the CARDAMOM inversion in 2014 was increased by 122%. Our results suggest an overestimated net spring flux in the JULES model and an overestimation of the net summer flux in both models of the biosphere. We propose that the difference seen between the prior and posterior flux estimates in summer and early autumn could be a result of the disturbance caused by crop harvest, leading to abrupt reduction in plant $CO_2$ uptake and increase in respiration sources, as it is not taken into account in either model. However, this hypothesis is just one of a combination of
uncertain factors that could lead to the differences seen, so further work would be needed to investigate the importance of crop harvest in UK $CO_2$ emissions.

The method developed and described here represents a first step towards looking at the UK biospheric $CO_2$ budget with a hierarchical Bayesian trans-dimensional MCMC inverse modelling framework. Further work is required to robustly constrain
biospheric $CO_2$ fluxes, through comparison with other model set-ups.

**6 Code availability**

Hierarchical Bayesian trans-dimensional MCMC code is available on request from Matthew Rigby (matt.rigby@bristol.ac.uk).

**7 Author contributions**

Emily White carried out the research. Emily White, Matt Rigby and Alistair Manning designed the research. Mark Lunt and
Anita Ganesan developed the model code. Simon O'Doherty, Kieran Stanley, Ann Stavert and Michel Ramonet provided data. Luke Smallman, Edward Comyn-Platt, Peter Levy and Mathew Williams provided model output. Emly White, Matt Rigby, Alistair Manning, Mark Lunt, Anita Ganesan, Simon O'Doherty, Kieran Stanley, Ann Stavert, Luke Smallman, Edward Comyn-Platt, Peter Levy and Paul Palmer wrote the text.



## 8 Acknowledgements

Emily White was supported by a studentship from the Natural Environment Research Council (NERC) Great Western 4+ Doctoral Training Partnership and NERC grant NE/M014851/1. Observations from BSD and HFD were supported under NERC grant NE/K002236/1. DECC network data are maintained by grant TRN1028/06/2015 from the UK Department of Business, Energy and Industrial Strategy. We are grateful to Gerry Spain and Victor Kazan for their work maintaining the measurements at MHD, as well as the support of RAMCES/ICOS, NUIG and EPA for this site. We are also grateful to Stephen

Humphrey and Andy MacDonald for their work maintaining the measurements at Tacolneston, and Carole Helfter and Neil Mullinger for their work maintaining the measurements at Bilsdale. Anita Ganesan was funded under a UK Natural Environment Research Council (NERC) Independent Research Fellowship (NE/L010992/1). T. L. Smallman was supported by NERC GHG program GREENHOUSE, grant NE/K002619/1. Paul Palmer gratefully acknowledged funding from NERC under grant reference NE/K002449/1, and gratefully acknowledges his Royal Society Wolfson Research Merit Award.

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



**Table 1:** Measurement site information.

| Site | Site code | Location | Inlet Height (m above ground level) | Network |
|---|---|---|---|---|
| Mace Head | MHD | 53.327 °N, 9.904 °W | 23 | LSCE |
| Ridge Hill | RGL | 51.998 °N, 2.540 °W | 90 | DECC |
| Tacolneston | TAC | 52.518 °N, 1.139 °E | 185 | DECC |
| Heathfield | HFD | 50.977 °N, 0.231 °E | 100 | GAUGE |
| Bilsdale | BSD | 54.359 °N, 1.150 °W | 248 | GAUGE |
| Angus | TTA | 56.555 °N, 2.986 °W | 222 | DECC |

**Table 2:** Probability density functions (PDFs) for parameter and hyper-parameter scaling factors. Mean and st. dev. in fourth and fifth columns relate to lognormal PDFs, lower bound and upper bound relate to uniform PDFs.

| Parameter | | PDF | Mean / lower bound | St. dev. / upper bound |
|---|---|---|---|---|
| **Prior uncertainty** | | | | |
| GPP | $x_{GPP}$ | Lognormal | 1 | 1 |
| | $\sigma_{x_{GPP}}$ | Uniform | 0.1 | 1.5 |
| Rtot | $x_{R_{tot}}$ | Lognormal | 1 | 1 |
| | $\sigma_{x_{R_{tot}}}$ | Uniform | 0.1 | 1.5 |
| Boundary conditions | $x_{BC}$ | Lognormal | 1 | 1 |
| | $\sigma_{x_{BC}}$ | Uniform | 0.01 | 0.05 |
| **Model-measurement representation uncertainty** | | | | |
| Standard deviation | $\sigma_y$ | Uniform | 0.3 ppm | 15 ppm |
| Correlation timescale | $\tau$ | Uniform | 1 hour | 120 hours |






**Table 3:** Specifications for different priors.

|  | Spatial Resolution | Temporal Resolution |
|---|---|---|
| **Biogenic fluxes** | | |
| JULES | $0.25° \times 0.25°$ | 2-hourly |
| CARDAMOM | 25 km × 25 km ($1° \times 1°$ outside the UK) | 2-hourly |
| **Anthropogenic fluxes** | | |
| NAEI (UK) | 1 km × 1 km | 2-hourly |
| EDGAR (outside UK) | $0.1° \times 0.1°$ | Yearly (using 2010) |
| **Ocean fluxes** | $4° \times 5°$ | Monthly (climatology) |


**Table 4:** Posterior UK estimates for the maximum net biospheric source and sink (values also shown in Fig. 5). The month in brackets indicates the month in which the maximum source/sink occurred.


|  | Year | Max. sink (Tg $CO_2$ yr$^{-1}$) | | Max. source (Tg $CO_2$ yr$^{-1}$) | |
|---|---|---|---|---|---|
| **CARDAMOM** | 2013 | $-298 \pm^{140}_{136}$ | (June) | $171 \pm^{94}_{76}$ | (January) |
|  | 2014 | $-360 \pm^{87}_{88}$ | (June) | $273 \pm^{65}_{63}$ | (November) |
| **JULES** | 2013 | $-456 \pm^{90}_{91}$ | (June) | $122 \pm^{83}_{78}$ | (December) |
|  | 2014 | $-542 \pm^{97}_{100}$ | (June) | $195 \pm^{65}_{70}$ | (October) |








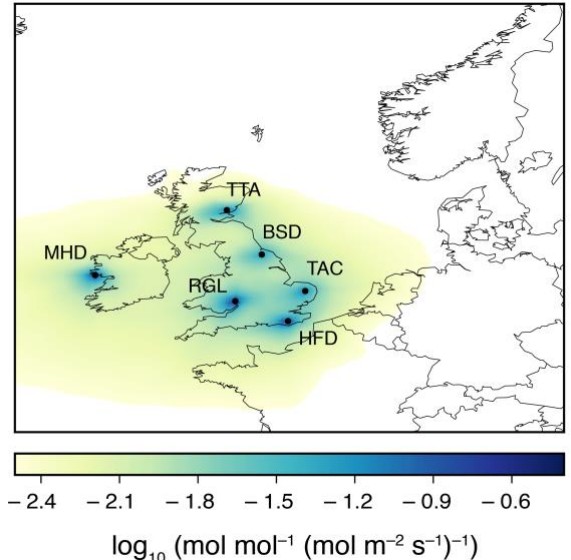

**Figure 1:** Mean annual NAME footprint for 2014, for each of the six sites. MHD: Mace Head; RGL: Ridge Hill; HFD: Heathfield; TAC:
Tacolneston; BSD: Bilsdale; TTA: Angus.





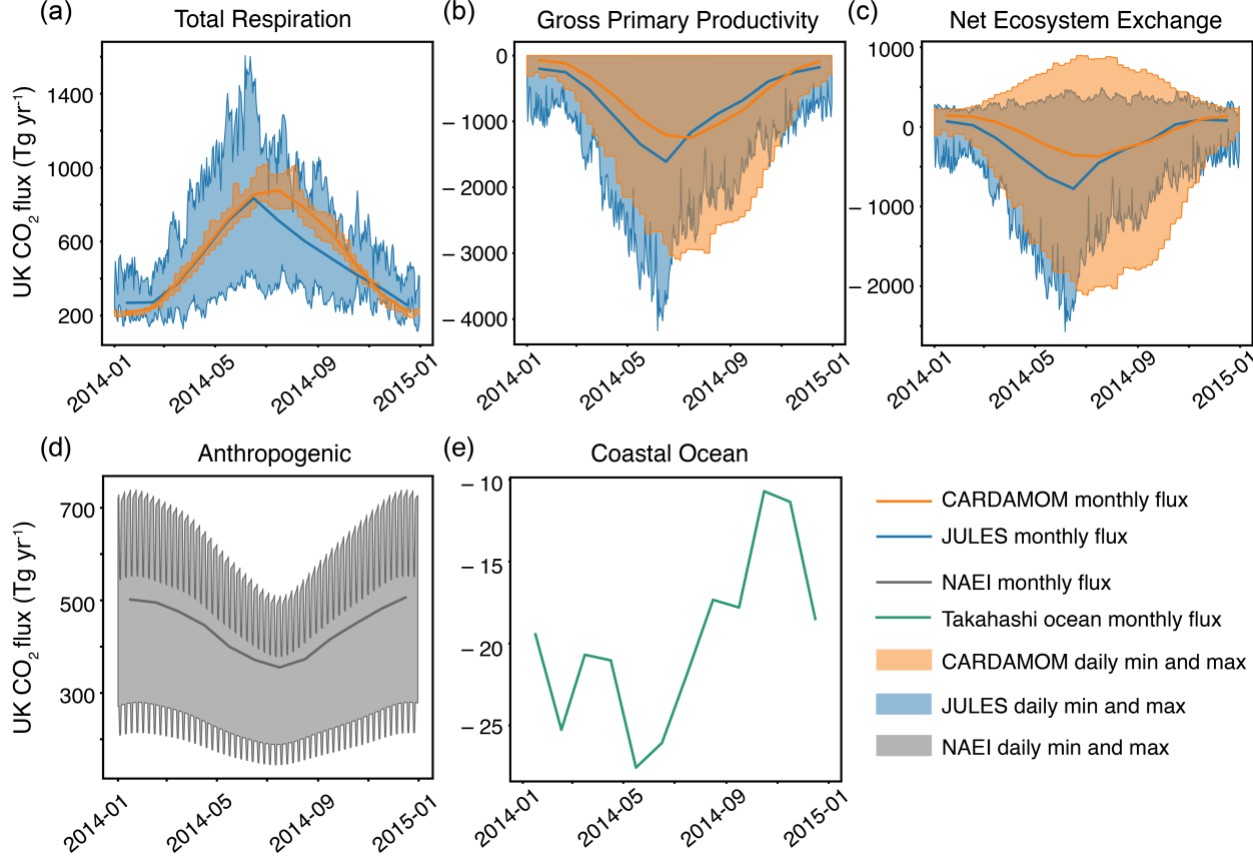

**Figure 2:** Prior UK fluxes in 2014. (a-c) Comparison of JULES (blue) and CARDAMOM (orange) monthly fluxes and minimum and
maximum daily values for TER, GPP and NEE respectively. (d) Monthly anthropogenic fluxes and minimum and maximum daily values
from the NAEI inventory within the UK. (e) Monthly coastal ocean net fluxes from the Takahashi et al. (2009) ocean CO$_2$ flux product.



**Figure 3:** Average prior flux maps for winter 2013 (December 2013, January – February 2014). (a) TER from CARDAMOM; (b) TER from JULES; (c) the difference between CARDAMOM and JULES TER; (d) GPP from CARDAMOM; (e) GPP from JULES; (f) the difference between CARDAMOM and JULES GPP; (g) NEE from CARDAMOM; (h) NEE from JULES; (i) the difference between CARDAMOM and JULES NEE.







**Figure 4:** Prior average flux maps for summer 2014 (June – August 2014). (a) TER from CARDAMOM; (b) TER from JULES; (c) the difference between CARDAMOM and JULES TER; (d) GPP from CARDAMOM; (e) GPP from JULES; (f) the difference between CARDAMOM and JULES GPP; (g) NEE from CARDAMOM; (h) NEE from JULES; (i) the difference between CARDAMOM and JULES NEE.





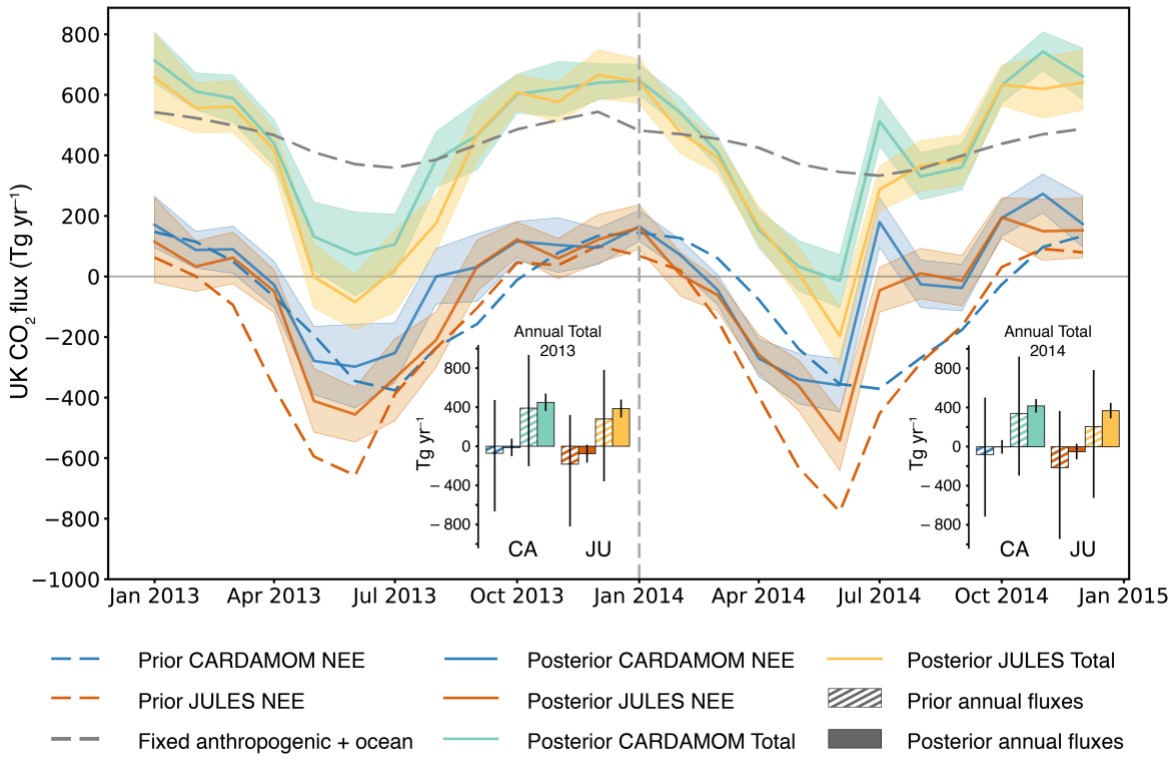

**Figure 5:** Posterior monthly net UK $CO_2$ flux (+ve is emission to atmosphere). Orange and blue monthly fluxes are posterior net biospheric (NEE) fluxes. Prior biosphere fluxes from JULES and CARDOMOM are shown in dashed orange and blue lines respectively. The fixed anthropogenic and ocean fluxes are denoted by the dark grey dashed line. Yellow and green monthly fluxes are the sum of the posterior NEE fluxes and the fixed anthropogenic and ocean fluxes. Shading represents $5^{th} - 95^{th}$ percentile. The bar charts represent annual net UK $CO_2$ flux for 2013 (left) and 2014 (right). Hashed bars denote prior annual fluxes, solid bars denote posterior annual fluxes. The bar colours correspond to the line colours: left hand bars for each model are NEE fluxes, right hand bars for each model are total fluxes (NEE + fixed sources). Uncertainty bars represent $5^{th} - 95^{th}$ percentile. CA – CARDAMOM. JU – JULES.





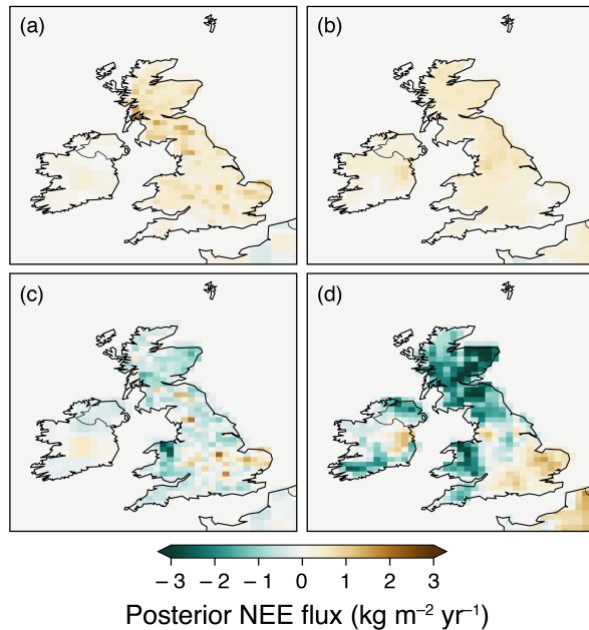

**Figure 6:** Posterior net biospheric (NEE) flux maps averaged over winter 2013 (December 2013, January – February 2014) and summer 2014 (June – August 2014). (a) Winter NEE flux from CARDAMOM inversion. (b) Winter NEE flux from JULES inversion. (c) Summer NEE flux from CARDAMOM inversion. (d) Summer NEE flux from JULES inversion.