# Peer review of "Quantifying the UK's Carbon Dioxide Flux: An atmospheric inverse modelling approach using a regional measurement network"

_Atmospheric Chemistry and Physics, 2018_

## Referee Comment (RC1) · Anonymous Referee #3 · 22 Oct 2018

**Quantifying the UK's Carbon Dioxide Flux: An atmospheric inverse modelling approach using a regional measurement network (White et al., 2018)**

**General comments**

The authors present a comprehensive quantitative study of the UK's $CO_2$ flux in 2013 and 2014, using an inverse approach that separately scales GPP and respiration. Their data assimilation system makes use of regional ground-based observations, two different models for the biospheric prior, and a LPDM for the atmospheric transport (which they modified to include the option of using time-disaggregated footprints). In addition to the net fluxes, the impact of the biospheric prior on the obtained posterior fluxes is studied in detail.

Overall the work is well written and of high quality. The manuscript provides sufficient details about the different components of the assimilation framework and contains an extensive discussion of the prior and posterior fluxes. To strengthen the presented conclusions, however, it would be good to pay some more attention to the technical details of the inversion results.

**Specific comments**

- On p.6, the authors assess the impact of the timespan for which footprints are disaggregated, and they conclude that the effect of going from 24h to 72h of time-disaggregation is negligible. However, I do not find the provided evidence for this statement very convincing. Indeed, the mole fractions calculated at Ridge Hill coincide nicely for 24h, 48h and 72h, but is this result representative for the entire UK? And how do the results compare to a simulation without time-disaggregation? The expected range of net annual biospheric fluxes has shifted significantly between the simulations with 24h of aggregation and with 72h of disaggregation, compared to the prior. It would be helpful if the results for no and for 48h of time-disaggregated footprints are included in table S1 as well. In addition, the uncertainty of the obtained net flux seems to be unaffected by the timespan of the disaggregated footprint. Can you comment on this?

- On p.12 the basis functions for the inversion are shortly discussed. This part could be made more clear by including a figure showing the clustering of the scaling factors.

- The inversion framework allows to scale the prior estimates for respiration and GPP separately. In a synthetic test it is shown that this approach indeed allows to compensate for biases in either respiration or GPP, which is obviously not possible with the NEE inversion. To strengthen the point of the superior behavior of the TER+GPP inversion over the NEE inversion, it would be nice to instead include a synthetic test with a truth that less obviously favors one approach over the other, e.g. by using a combination of the JULES and CARDAMOM fluxes as truth.

- A different anthropogenic flux map (EDGAR) is used outside the UK. Are these fluxes in agreement with the NAEI fluxes for the UK?

- The data statistics in table S2 cover the year 2014 only. Are the statistics for 2013 similar, and why are they not included? It would be nice to see these statistics for the entire range of presented inversion results, i.e. both 2013 and 2014. Moreover, for the inversion with JULES prior the bias increases for 2 out of 6 measurement

sites. Can you comment on this?

In addition, I find it peculiar that the prior RMSE and prior bias differ between the GPP-TER and the NEE inversion. How did you evaluate these statistics?

I would also suggest to include the statistics for the NEE-inversion in table S2 (maybe put them in a smaller font and/or grey) and to move the table to the main paper.

- The optimized state vector and hyper-parameters are not discussed, while it would be interesting to have at least a look at the number of time periods resolved in the inversion, how scaling factors for GPP and TER vary, and to what extent the boundary conditions required scaling during the inversion. Please include such a discussion.

- Due to the use of satellite remote-sensing data in CARDAMOM, one would expect a benefit of using this prior over the JULES model, at least with respect to spatial distribution of the fluxes. This point does not seem to be addressed.

- Figure 6 can be improved by adding two additional subfigures that show the difference between CARDAMOM and JULES posterior fluxes. Also the addition of maps showing the related uncertainty in posterior NEE would be helpful.

**Technical corrections**

- P.2, the abbreviation GHG is used before it is introduced.

- P.5, please include an indication of the uncertainty related to the $CO_2$ measurements

- P.10, please mention the spatiotemporal resolution of the prior ocean flux estimates.

- P.11, to reduce initial confusion, it might be useful to move the paragraph starting at line 350 (about the variable time instead of space dimension) to before equation 4.

- Please be consistent in the colors you use: e.g. the colors for CARDAMOM and JULES are reversed between figures 2 and 5.

- The section numbering in chapter 4 should be updated, now 4.2 comes after 4.3

---

## Referee Comment (RC2) · Anonymous Referee #1 · 23 Oct 2018

The manuscript "Quantifying the UK's Carbon Dioxide Flux: An atmospheric inverse modelling approach using a regional measurement network" by E. White et al. presents an estimation of the UK's net $CO_2$ fluxes over two years based on an atmospheric inverse modelling approach and a measurement network around and within the UK. They find that averaged over the two years the UK's annual net biosphere flux is close to zero, i.e. in balance, within the error bars.

The research in itself, i.e. atmospheric inverse modelling, is not new, however the focus on national scale is somewhat new and has raised considerable interest in the recent past because of the growing importance of national greenhouse gas reporting.

[Figure]

Since the reporting is based on bottom-up methods, inverse modellings as a top-down approach can be considered is a tool to evaluate the reporting. Another interesting aspect of the paper is the sensitivity study with respect to the underlying prior flux field and the approach to solve for gross fluxes instead of the net flux in the inversion. So, overall the manuscript addresses an important issue in the field of carbon cycle research linking to atmospheric measurements.

The manuscript is well written and structured and mostly easy to read and follow. There are no major revisions needed, however, a few minor points need to be addressed before the manuscript can be published.

A main point of critique is a missing validation or at least evaluation of the inversion results and posterior fluxes. This is of course not an easy task but at least some basic evaluation tests should have been performed. This could be done by comparing modelled $CO_2$ vertical profiles using the posterior fluxes against aircraft measurements or, if not available, ground-based observations withheld from the inversion. Also, the resolution of the posterior fluxes might already be high enough to compare them directly with eddy covariance based observations. Such an evaluation is missing completely. It is therefore not clear how 'trustworthy' the posterior fluxes are and, also, which one of the two inversions based on different priors performs better than the other. This is a crucial point currently missing in the manuscript and should be added before publication.

Some additional points: L 28: Spell out negative.

L 43: Are flux measurements really localized down to centimetres? Probably not.

L 46: What do you mean by 'are driven by observational data to varying degree of detail'?

L 55/56: Indeed, inversions are a valuable tool, but they are also not free from errors. It would be good to mention here sources for uncertainties in inversions and put these

into perspective.

L 62: This is of course not true: Using these measurements in an inversion framework is not an independent way of providing estimates GHG emissions because the inverse modelling system requires prior emission fields as an input. Hence it is not independent of bottom-up inventories. This sentence needs to be rephrased in the manuscript.

L 74: Why is it rarely the case that model uncertainties are well characterized? This is also related to the comment on L 55/56.

L 80/81: But using Gaussian PDFs is only a choice made by the user, there is no mathematical need for Gaussian PDFs, one can use any PDF to describe prior knowledge. So why is that a problem here?

L 83/84: Why is the size of the diurnal cycle a problem and how does it matter if you solve for monthly fluxes?

L 156/157: Shouldn't this 'surface-exchange' height be dependent of actual meteorological conditions and vary for instance with boundary layer height or the strength of vertical mixing?

Sec 2.2.2: Have you done some sensitivity tests on how to handle the boundary conditions? It would be interesting to see how the results change if you don't include the boundary conditions in the control vector. There is of course a trade off between getting the boundary conditions right and using as much of the observational information as possible to constrain the surface fluxes. In principle, the boundary conditions are nuisance variables, which obviously influence your results but are in themselves not very interesting.

L 235 and Sections 2.3.2 and 2.3.3: The wording is maybe a bit misleading here. First you say, that ocean and anthropogenic fluxes are subtracted, and thus treated as perfectly known. Then you explain that these are prior fluxes. Usually a prior flux is a flux that gets updated through the inversion yielding a posterior flux. But you do not do

that here. I suggest to reword Sections 2.3.2 and 2.3.3 and not use the word 'prior' for ocean and anthropogenic fluxes. Also, I wonder how well the ocean fluxes are really know in that area, especially if you take the Takahashi fluxes (representing open ocean fluxes) as an estimate of the UK coastal ocean fluxes!

Ll 293-295: Does that mean that only MODIS LAI is assimilated? That also means that you are assimilating model output (since MODIS LAI is not a measured or even observed quantity).

L 303: It seems that both biosphere models use MODIS LAI data in some way. How independent are then the estimates of JULES and CARDAMOM?

L 364: How did you determine the length of the burn-in period and does the number of iterations include the burn-in period?

Sec 2.4.3 and Eq (7): What is x and y here? How many basisfunctions do you have in total and how does the Jacobian look like? Maybe you can add an equation for the Jacobian: H= del . . ./ del . . .

L 393: The word 'tested' is not correct here, 'applied' would fit better. In any case it would be good to add a few sentences on testing you set up in e.g. an identical twin experiment.

L 414: How can soil and litter carbon stocks be fixed in JULES? I wonder a model with fixed carbon stocks can provide decent estimates of the actual respiration fluxes.

Section 3.1: This goes back to my main comment on evaluating the inversions. Can you say which result is more realistic?

L 480: Do you mean 'underestimating' the net summer flux compared to the true flux? And if so, how do you know the true flux?

L 516: Do you mean here the posterior fluxes from the inversions or the prior fluxes from the two different models? Maybe stick to a common notation/terminology for the

fluxes, e.g. prior fluxes and posterior fluxes throughout the manuscript and not refer to them just by model name.

Sec 3.4: This section presents some posterior diagnostics of the inversions and presents a first step towards an evaluation of the inversions. What do the different fits to the data mean for the inversions and posterior fluxes?

Ll 588-590: Agricultural activities should somehow (implicitly) be accounted for by the biosphere models through the use of MODIS LAI, which should capture events such as harvest in the LAI.

Ll 622-626: Again, this hypothesis means that you trust your inversion results but it is not clear on which basis you trust the inversions. This hypothesis should be supported by a more substantial evaluation of the inversions.

---

## Author Response (AR1)

We are very grateful for the helpful comments of the referees, which we have used to improve the manuscript as described below.

Please note:
In the original text the CARDAMOM-DALEC modelling framework was referred to as CARDAMOM, however to avoid confusion about how this framework works it is now referred to as DALEC in the manuscript. Therefore, the model name has been changed to DALEC in the author comments and DALEC is used in the responses below.

Anonymous Referee #1

Reviewer comment:
A main point of critique is a missing validation or at least evaluation of the inversion results and posterior fluxes. This is of course not an easy task but at least some basic evaluation tests should have been performed. This could be done by comparing modelled $CO_2$ vertical profiles using the posterior fluxes against aircraft measurements or, if not available, ground-based observations withheld from the inversion. Also, the resolution of the posterior fluxes might already be high enough to compare them directly with eddy covariance based observations. Such an evaluation is missing completely. It is therefore not clear how 'trustworthy' the posterior fluxes are and, also, which one of the two inversions based on different priors performs better than the other. This is a crucial point currently missing in the manuscript and should be added before publication.

Author response:
The posterior fluxes have been used to model mole fractions at Weybourne. Table 5 (Table 1 in response supplement PDF) includes the fit to the data of modelled mole fractions using prior and posterior fluxes for all 4 (DALEC, JULES, gross flux, net flux) inversions for the WAO site. Fig. S14 (Fig. 1 in response) shows the residual mole fractions for all 4 inversions. Fit to the data is improved using the posterior vs prior fluxes.

Change to manuscript:
To test our posterior results against data that has not been included in the inversion, the posterior fluxes have been used to simulate mole fractions at Weybourne Atmospheric Observatory (see Fig. 1 for location in relation to the other sites and Table 1 for site information). The statistics of fit to the data are given in italics in Table 5 and show an improvement in $R^2$ of 0.18 with the DALEC inversion and 0.13 with the JULES inversion, an improvement in RMSE of 1.09 ppm with the DALEC inversion and 0.75 ppm with the JULES inversion and an improvement in the mean bias of 0.64 ppm in the DALEC inversion and 0.56 in the JULES inversion. These results show that the a posteriori fluxes improve the fit to the data at a measurement station not included in the inversion. The results are very similar between the two inversions at this site, but suggest that the DALEC inversion may perform slightly better, at least in this region of the UK. Figure S14 shows the residual mole fractions at Weybourne for each of the inversions carried out in this work.

Reviewer comment:
L 43: Are flux measurements really localized down to centimetres? Probably not.

Author response:
Agreed, changed to metres.

Change to manuscript:
However, they are relatively localised estimates (metres to hectares), which are challenging to scale up to national levels.

Reviewer comment:
L 46: What do you mean by 'are driven by observational data to varying degree of detail'?

Author response:
Here we were trying to convey that different observation-based products may be assimilated into biosphere models at different temporal and spatial resolutions. Wording is changed to clarify.

Change to manuscript:
Such models describe processes to varying degrees of complexity, with poorly described errors and are driven by observational data at differing temporal and spatial resolutions; hence predictions of biogenic greenhouse gas (GHG) fluxes have poorly quantified biases and can vary significantly between models (Todd-Brown et al., 2013; Atkin et al., 2015).

Reviewer comment:
L 55/56: Indeed, inversions are a valuable tool, but they are also not free from errors. It would be good to mention here sources for uncertainties in inversions and put these into perspective.

Author response:
Added the text below

Change to manuscript:
However, errors in atmospheric transport, unknown uncertainties related to the prior fluxes and issues surrounding the under-determined nature of the problem are all limitations of this approach.

Reviewer comment:
L 62: This is of course not true: Using these measurements in an inversion framework is not an independent way of providing estimates GHG emissions because the inverse modelling system requires prior emission fields as an input. Hence it is not independent of bottom-up inventories. This sentence needs to be rephrased in the manuscript.

Author response:
Rephrased as

Change to manuscript:
To support this legislation, a continuous and automated measurement network has been established (Stanley et al., 2018; Stavert et al., 2018) with the goal of providing estimates of GHG emissions using methods that are complementary to those used to compile the UK's bottom-up emissions inventory, reported annually to the UNFCCC.

Reviewer comment:
L 74: Why is it rarely the case that model uncertainties are well characterized? This is also related to the comment on L 55/56.

Author response:
Explanation has been added

Change to manuscript:

In practice, this is rarely the case because, for example, uncertainties related to the atmospheric transport model are poorly understood and uncertainties related to biospheric flux estimates from models are largely unknown.

Reviewer comment:
L 80/81: But using Gaussian PDFs is only a choice made by the user, there is no mathematical need for Gaussian PDFs, one can use any PDF to describe prior knowledge. So why is that a problem here?

Author response:
In an analytical inversion there is a mathematical need for Gaussian PDFs and the majority of previous studies will use Gaussian PDFs. Very few existing set-ups can accommodate non- Gaussian PDFs.

Change to manuscript:
Furthermore, for reasons of mathematical and computational convenience, Gaussian probability density functions (PDFs) are commonly used to describe prior knowledge (e.g. Miller et al., 2014).

Reviewer comment:
L 83/84: Why is the size of the diurnal cycle a problem and how does it matter if you solve for monthly fluxes?

Author response:
Being able to simulate the diurnal cycle of the observations as closely as possible is very important if you want to make the most of high frequency data. Figure S2 (Fig. 2 in response) has been adapted in the revised manuscript and shows what the simulated mole fractions look like if monthly fluxes are used. They are much smaller in magnitude and 12 hours out of phase with the data. Table S1 (Table 2 in response supplement PDF) shows an inversion result using monthly fluxes and it is not at all realistic. Therefore, not taking the diurnal cycle into account has a large impact on the inversion.

Change to manuscript:
The choice of 24-hour disaggregation balanced considerations of computational efficiency and simulation accuracy. For certain months and sites, we carried out a set of tests to determine how sensitive our simulated mole fractions and inversion results were, when footprints were disaggregated for the first 12 to 72 hours prior to each measurement (Figure S2; Table S1). Assuming that the 72-hour simulations were the most accurate, we found little degradation in performance by using only 48 or 24 hours disaggregation, when compared to the other uncertainties in the system (e.g. differences between fluxes derived using the 24, 48 and 72 hour simulations were smaller than the 90% confidence interval). However, when only 12 hours was used (or fully integrated footprints), the modelled diurnal cycle was out of phase with the observations.

Reviewer comment:
L 156/157: Shouldn't this 'surface-exchange' height be dependent of actual meteorological conditions and vary for instance with boundary layer height or the strength of vertical mixing?

Author response:

Our team has tested various different schemes (e.g. different "surface heights" using with similar inversion frameworks in Manning et al., 2011 and Arnold et al., 2018), and results seem to be relatively insensitive to this choice.

Reviewer comment:
Sec 2.2.2: Have you done some sensitivity tests on how to handle the boundary conditions? It would be interesting to see how the results change if you don't include the boundary conditions in the control vector. There is of course a trade off between getting the boundary conditions right and using as much of the observational information as possible to constrain the surface fluxes. In principle, the boundary conditions are nuisance variables, which obviously influence your results but are in themselves not very interesting.

Author response:
We have tested this by carrying out the DALEC inversion with boundary conditions that have been perturbed by +/- 1ppm a priori. The results are shown in Table 3 in the response supplement PDF and a summary of the test has been added to the text.

Change to manuscript:
A sensitivity test where 1ppm is added or taken away from the mole fractions at the domain edges indicates that in June a $\pm$1ppm change translates to a 1-3% change in the inversion result and in December a $\pm$1ppm change translates to a 7-11% change in the inversion result. These changes are substantially smaller than the posterior uncertainty.

Reviewer comment:
L 235 and Sections 2.3.2 and 2.3.3: The wording is maybe a bit misleading here. First you say, that ocean and anthropogenic fluxes are subtracted, and thus treated as perfectly known. Then you explain that these are prior fluxes. Usually a prior flux is a flux that gets updated through the inversion yielding a posterior flux. But you do not do that here. I suggest to reword Sections 2.3.2 and 2.3.3 and not use the word 'prior' for ocean and anthropogenic fluxes. Also, I wonder how well the ocean fluxes are really know in that area, especially if you take the Takahashi fluxes (representing open ocean fluxes) as an estimate of the UK coastal ocean fluxes!

Author response:
Changed wording around anthropogenic and ocean fluxes. The ocean component is small so the ocean fluxes do not have a significant impact on the results.

Change to manuscript:
Since the oceanic flux component is small, the comparatively low temporal and spatial resolution of these flux estimates does not significantly impact the inversion results.

Reviewer comment:
L 293-295: Does that mean that only MODIS LAI is assimilated? That also means that you are assimilating model output (since MODIS LAI is not a measured or even observed quantity).

Author response:
It means that MODIS LAI, a temporally and spatially explicit estimate of biomass and a spatially explicit estimate of soil carbon are assimilated. Yes, these are model outputs and the wording has been changed to clarify this.

Change to manuscript:

Observation-derived information used in the current analysis are satellite-based remotely sensed time series of Leaf Area Index (LAI) (MODIS; MOD15A2 LAI-8 day version 5, http://lpdacc.usgs.gov/), a prior estimate of above ground biomass (Thurner et al., 2014) and a prior estimate of soil organic matter (Hiederer and Köchy, 2012).

Reviewer comment:
L 303: It seems that both biosphere models use MODIS LAI data in some way. How independent are then the estimates of JULES and DALEC?

Author response:
There are two key differences between the models in terms of LAI. Firstly, DALEC uses temporally explicit estimates whereas JULES uses a climatology, which means that DALEC is able to capture the interannual variability. Secondly, DALEC assimilates MODIS LAI within a calibration process to simulate its own LAI. This means that DALEC can shift away from MODIS LAI estimates which, through combination with other data, the framework finds to be unlikely. Aside from these points, the models also have very different structures (physiological representation) and parameterisation (ecosystem traits), for example very different descriptions of the carbon assimilation and respiration mechanisms, so they would give different estimates of GPP for the same LAI anyway. We have reworded the description of DALEC and the paragraph about the differences between the models to clarify these points.

[revised manuscript text omitted]

Reviewer comment:
L 364: How did you determine the length of the burn-in period and does the number of iterations include the burn-in period?

Author response:
The number of iterations does not include the burn-in period. This has been clarified in the text. The length of the burn-in was determined by visual inspection of the chains and a conservatively large estimate was used to ensure that it was sufficient.

Change to manuscript:
The algorithm had a burn-in period of $5×10^4$ iterations and was then run for an additional $2×10^5$ iterations to appropriately explore the posterior distribution.

Reviewer comment:

Sec 2.4.3 and Eq (7): What is x and y here? How many basis functions do you have in total and how does the Jacobian look like? Maybe you can add an equation for the Jacobian: H= del .../ del ...

Author response:
X and y have been described already in Eq. 3. There are 19 spatial basis functions, this has been clarified in the text. Parts of Section 2.4.3 about the Jacobian matrix have been reworded to clarify how the Jacobian is set up.

Change to manuscript:
A scaling factor is solved in the inversion, scaling GPP and TER within the 4 outer regions and within maps of five or six PFTs in the sub-domain: broadleaf tree, needleleaf tree, C3 grasses, C4 grasses, shrubland and, in the case of TER, bare soil. Therefore, there are 19 spatial basis functions.

H has dimensions m (number of data points) by n (number of parameters).

To create this linear model, we multiplied the footprints by the prior GPP and TER fluxes separately, then multiplied these by the fractional map of basis functions (described in Sect. 2.4.2) and summed over the domain. The boundary conditions were broken down by four further basis functions for each edge of the domain as explained in Sect. 2.2.2. The parameters vector, x, consisted of a set of scaling factors that multiplied the fluxes or boundary conditions. Multiplying the sensitivity matrix by the prior estimate of x, a vector of ones, yields the prior modelled mole fraction time-series at a site.

Reviewer comment:
L 393: The word 'tested' is not correct here, 'applied' would fit better. In any case it would be good to add a few sentences on testing you set up in e.g. an identical twin experiment.

Author response:
We propose not to change the wording. We feel that the presented synthetic tests and agreement between two inversions shows sufficient evidence that our system performs well. Furthermore, many more synthetic tests have been carried out, which are too numerous to show here, but will be citable by the time of publication.

Reviewer comment:
Section 3.1: This goes back to my main comment on evaluating the inversions. Can you say which result is more realistic?

Author response:
From the validation study, DALEC performs marginally better than JULES but not significantly, therefore it is difficult to say which is best. Some suggestion of this has been added to Section 3.4:

Change to manuscript:
[About the fits to data included in the inversion]
Overall, the fits are relatively similar between the DALEC and JULES inversions implying that the two inversions perform similarly well.

[About the fits to the validation data set]
Again, the results are very similar between the two inversions but suggest that the DALEC inversion may perform slightly better, at least in this region of the UK.

Reviewer comment:
L 414: How can soil and litter carbon stocks be fixed in JULES? I wonder a model with fixed carbon stocks can provide decent estimates of the actual respiration fluxes.

Author response:
The main driver of temporal variability in soil respiration in JULES is the soil temperature given the Q10 and the second largest driver is soil moisture. It is true that an element of temporal variability is lost using a fixed soil carbon. However, this would be the second or third most important driver of temporal variability. Also, there is a lack of such observations of this behaviour (e.g. changes in soil carbon stocks over time driving changes in soil respiration), hence using a fixed map, which gives a reasonable spatial distribution, was the preferred option here.

Reviewer comment:
L 480: Do you mean 'underestimating' the net summer flux compared to the true flux? And if so, how do you know the true flux?

Author response:
We meant compared to the posterior flux.

Change to manuscript:
Generally, the models underestimate the net summer flux compared to the posterior flux (to the greatest extent in 2014), although the summer estimate from the JULES inversion in 2013 is not statistically different from the prior.

Reviewer comment:
L 516: Do you mean here the posterior fluxes from the inversions or the prior fluxes from the two different models? Maybe stick to a common notation/terminology for the fluxes, e.g. prior fluxes and posterior fluxes throughout the manuscript and not refer to them just by model name.

Author response:
We meant posterior fluxes from the inversions. The sentence has been rewritten.

Change to manuscript:
However, the net sink in the JULES inversion is larger than the DALEC inversion in Scotland, south Wales, Northern Ireland and south-west England.

Reviewer comment:
Sec 3.4: This section presents some posterior diagnostics of the inversions and presents a first step towards an evaluation of the inversions. What do the different fits to the data mean for the inversions and posterior fluxes?

Author response:
Added more detail to Section 3.4

Change to manuscript:
Agreement between the data and the posterior simulated mole fractions at the measurement sites used to constrain the inversion is greatly improved compared to prior simulated mole fractions, with R2 values increasing by a minimum of 0.24 and up to 0.5 (to give values ranging between 0.53 and 0.71) and root mean square error (RMSE) decreasing by at least

1.35 ppm and up to 2.6 ppm (to give values ranging between 1.26 ppm and 2.71 ppm). Table 5 shows all statistics for the prior and posterior mole fractions compared to the observations of atmospheric $CO_2$ concentrations. Overall, the fits are relatively similar between the DALEC and JULES inversions, implying that the two inversions perform similarly well by these metrics. In terms of $R^2$, the best fit to the data is observed at Heathfield in the DALEC inversion and Angus in the JULES inversion. In terms of RMSE, the best fit to the data is observed at Angus in the DALEC inversion and Mace Head in the JULES inversion. The smallest posterior mean bias is observed at Angus in the DALEC inversion and Ridge Hill in the JULES inversion. Therefore, there are some small spatial differences in how well each of the inversions is able to fit the data but no clear indication of which areas of posterior flux might be subject to the largest improvement in either inversion. Figures S12 and S13 show the residual mole fractions in 2014 and indicate that residuals are somewhat larger during the summer than the winter

Reviewer comment:
L 588-590: Agricultural activities should somehow (implicitly) be accounted for by the biosphere models through the use of MODIS LAI, which should capture events such as harvest in the LAI.

Author response:
There are a couple of reasons why MODIS LAI is not able to capture events such as harvest.
1) MODIS LAI uses constant reflectance fractions so even though crops will have a different reflectance fraction to forest, for example, the fact that surface reflectance of crops change dramatically with senescence and post-harvest litter on the soil surface is not accounted for. This will partially mean that there will be errors introduced into MODIS's LAI estimates for crops.
2) Even with the correct LAI, crops are different from other ecosystems in terms of their C flux due to the removal of biomass which would otherwise decompose. This is in addition to harvest itself introducing litter into the system, which decomposes at a different time than it would if it was "naturally" occurring. In neither DALEC or JULES was a model structure which allows for C redistribution within the ecosystem / or harvest removal processes represented which means that the models cannot reproduce the associated fluxes.
This is therefore another area where performing an inversion can help to resolve these features that are hard to identify with MODIS LAI.

Reviewer comment:
L 622-626: Again, this hypothesis means that you trust your inversion results but it is not clear on which basis you trust the inversions. This hypothesis should be supported by a more substantial evaluation of the inversions.

Author response:
Addressed in Anonymous Referee #1's first comment.

Anonymous Referee #3

Reviewer comment:
On p.6, the authors assess the impact of the timespan for which footprints are disaggregated, and they conclude that the effect of going from 24h to 72h of time-disaggregation is negligible. However, I do not find the provided evidence for this statement very convincing. Indeed, the mole fractions calculated at Ridge Hill coincide nicely for 24h,

48h and 72h, but is this result representative for the entire UK? And how do the results compare to a simulation without time-disaggregation? The expected range of net annual biospheric fluxes has shifted significantly between the simulations with 24h of aggregation and with 72h of disaggregation, compared to the prior. It would be helpful if the results for no and for 48h of time- disaggregated footprints are included in table S1 as well. In addition, the uncertainty of the obtained net flux seems to be unaffected by the timespan of the disaggregated footprint. Can you comment on this?

Author response:
Figure S2 (Fig. 2 in response) has now been replotted, showing the results for no time-disaggregation using monthly fluxes, 6 hours back and 12 hours back, as well as the original 24, 48 and 72 hours. Fig. S2 (Fig. 2 in response) now also shows the modelled mole fractions at Tacolneston, showing the same result, indicating that it is representative of multiple stations in the UK. Table S1 (Table 2 in response supplement PDF) now includes results with no time disaggregation, along with disaggregated footprints going 12, 24, 48 and 72 hours back. Although there are differences between 24, 48 and 72 hours back, these are not statistically significant (within the 90%ile range). However, for 12 hours or inversions using the integrated footprints, the inversions are statistically different from the time-disaggregated results. Compared to the seasonal cycle or the differences between the JULES and DALEC inversions in Fig. 5, the differences between 24, 48 and 72 are small. The uncertainty reduction will primarily depend on the magnitude of the sensitivity (in addition to the number of measurements and the measurement/model uncertainty). Since we examine the net flux here, it is consistent that time-integrated footprint and the time-disaggregated footprints lead to similar uncertainty reduction in the net flux.

As an aside, the inversion result for the 72 hours back test has changed slightly. In the previous version, we carried out the 72 hours back inversion using data that had the anthropogenic and ocean forward modelled using 72 hours back footprints. However, we have now changed this for consistency, so all of these tests use data with fixed components removed using 24 hours back footprints.

Change to manuscript:
The choice of 24-hour disaggregation balanced considerations of computational efficiency and simulation accuracy. For certain months and sites, we carried out a set of tests to determine how sensitive our simulated mole fractions and inversion results were, when footprints were disaggregated for the first 12 to 72 hours prior to each measurement (Figure S2; Table S1). Assuming that the 72-hour simulations were the most accurate, we found little degradation in performance by using only 48 or 24 hours disaggregation, when compared to the other uncertainties in the system (e.g. differences between fluxes derived using the 24, 48 and 72 hour simulations were smaller than the 90% confidence interval). However, when only 12 hours was used (or fully integrated footprints), the modelled diurnal cycle was out of phase with the observations.

Reviewer comment:
On p.12 the basis functions for the inversion are shortly discussed. This part could be made more clear by including a figure showing the clustering of the scaling factors.

Author response:
We have added a figure (Fig. S6 – Fig. 3 in response) showing the spatial basis functions used in the inversion (based on PFTs). We hope this clarifies this discussion.

Change to manuscript:

Within the West-Central Europe area (the hatched region in Fig. S1), a map of the fraction of different plant functional types (PFTs) in each grid cell has been used to further break down the region (Fig. S6).

Reviewer comment:
The inversion framework allows to scale the prior estimates for respiration and GPP separately. In a synthetic test it is shown that this approach indeed allows to compensate for biases in either respiration or GPP, which is obviously not possible with the NEE inversion. To strengthen the point of the superior behavior of the TER+GPP inversion over the NEE inversion, it would be nice to instead include a synthetic test with a truth that less obviously favors one approach over the other, e.g. by using a combination of the JULES and DALEC fluxes as truth.

Author response:
To address the reviewer's comment, the synthetic test in Fig. S5 (Fig. 4 in response) has been changed to use synthetic data created using the DALEC biosphere fluxes, and the NAME model, while JULES is used for the prior fluxes.

Change to manuscript:
To demonstrate this, we have carried out a synthetic test (Fig. S5) in which we have investigated the ability of our inversion system to solve for a "true" flux, created using the DALEC prior fluxes and NAME simulations, in an inversion that used the JULES fluxes as the prior. Figure S5(a) shows that monthly posterior fluxes for the inversion where GPP and TER are separated agree with the "true" flux within estimated uncertainties in 16 out of 24 months. In contrast, whilst the posterior fluxes for the inversion where NEE is scaled has changed significantly from the prior, it is not in agreement with the "true" flux except in July 2013 and August and September 2014. The posterior diurnal cycles of GPP, TER and NEE, which are shown as an average for June 2014 in Fig. S5(b) and Fig. S5(c), highlight the differences in diurnal cycle between the two models. The inversion that can adjust the two sources separately leads to higher night-time fluxes, which are closer to the "true" flux than the prior. On the other hand, the inversion where NEE is scaled can only stretch or shrink the diurnal cycle in one direction, increasing both the daytime sink and night-time source, or decreasing them, together. In this case, they have decreased, which does bring the net June 2014 flux in Fig. S5(a) closer to the "true" June 2014 flux but cannot go far enough to reconcile the monthly fluxes.

Reviewer comment:
A different anthropogenic flux map (EDGAR) is used outside the UK. Are these fluxes in agreement with the NAEI fluxes for the UK?

Author response:
The fluxes are not in agreement (NAEI is 482 Tg/yr over the UK in 2013, 439 Tg/y in 2014, whilst EDGAR is 540 Tg/yr over UK in both years as we use the 2010 values throughout the inversion). However, our tests show that this does not have a significant impact on the derived fluxes from the UK.

Change to manuscript:
Within the UK, the NAEI and EDGAR fluxes differ by around 15% (540 Tg/yr for EDGAR, 460 Tg/yr for NAEI). We do not find that our derived UK fluxes are significantly affected by perturbations of this magnitude applied to anthropogenic emissions outside the UK.

Reviewer comment:

The data statistics in table S2 cover the year 2014 only. Are the statistics for 2013 similar, and why are they not included? It would be nice to see these statistics for the entire range of presented inversion results, i.e. both 2013 and 2014. Moreover, for the inversion with JULES prior the bias increases for 2 out of 6 measurement sites. Can you comment on this? In addition, I find it peculiar that the prior RMSE and prior bias differ between the GPP-TER and the NEE inversion. How did you evaluate these statistics? I would also suggest to include the statistics for the NEE-inversion in table S2 (maybe put them in a smaller font and/or grey) and to move the table to the main paper.

Author response:
This table has been changed to give statistics calculated across all months of the inversion. The table has also been merged with the table of statistics from the NEE inversion and moved to the main paper (Table 5 – Table 1 in response supplement PDF). The reason why the JULES prior bias increased for some sites and for the prior statistics to be different between the two inversions, is because we initially calculated the statistics for the prior using the posterior baseline (we did this because the prior mole fractions were strongly influenced by biases in the baseline mole fractions, however, we now realise that this was confusing, as it made it look as though the inversion could be making the fit to the data worse, which of course is not possible). We have now changed this to use the prior baseline to avoid confusion. Therefore, the plots of residual mole fractions Figs. S12-13 (Figs 5-6 in response) have also been updated to use the prior baseline with prior modelled mole fractions.

Change to manuscript:
Agreement between the data and the posterior simulated mole fractions at the measurement sites used to constrain the inversion is greatly improved compared to prior simulated mole fractions, with R2 values increasing by a minimum of 0.24 and up to 0.5 (to give values ranging between 0.53 and 0.71) and root mean square error (RMSE) decreasing by at least 1.35 ppm and up to 2.6 ppm (to give values ranging between 1.26 ppm and 2.71 ppm). Table 5 shows all statistics for the prior and posterior mole fractions compared to the observations of atmospheric CO2 concentrations.

Reviewer comment:
The optimized state vector and hyper-parameters are not discussed, while it would be interesting to have at least a look at the number of time periods resolved in the inversion, how scaling factors for GPP and TER vary, and to what extent the boundary conditions required scaling during the inversion. Please include such a discussion.

Author response:
A figure has been added to the supplement (Fig. S11 – Fig. 7 in response) showing the mean and maximum number of time periods that each month is broken down into for each spatial basis function for each month. The lines below have been added to the main text. In terms of the scaling factors, we do not think it is so useful to look at the scaling factors and hyperparameters in isolation and the figures showing the posterior fluxes for GPP and TER (Figs. S7-S8) already show this information in a more meaningful way.

Change to manuscript:
Fig. S11 shows that the inversion typically scaled the fluxes within 4 or 5 temporal regions per month, although for some parameters in some months scaling factors were found up to roughly a daily resolution.

Reviewer comment:

Due to the use of satellite remote-sensing data in DALEC, one would expect a benefit of using this prior over the JULES model, at least with respect to spatial distribution of the fluxes. This point does not seem to be addressed.

Author response:
In fact, both models use remote sensing LAI data from MODIS and the main difference in their use of this data is not in the spatial distribution but that DALEC uses a time-varying product with interannual variablilty and JULES uses a climatology. This does potentially give DALEC an advantage, however DALEC could also suffer due to the greater noise in the MODIS estimates which will be averaged out in the climatology. It is difficult to say what impact this has on the inversion compared to the other differences between the models. Importantly, the inversion helps to bring the two estimates together, despite the possible advantages of using DALEC. Some clarification has been added to the introduction and to Section 3.1

Change to manuscript:
[Introduction]
Gross primary productivity (GPP) and terrestrial ecosystem respiration (TER) estimates from the Joint UK Land Environment Simulator (JULES) and Data Assimilation Linked Ecosystem Carbon (DALEC) are used as prior flux constraints. JULES is a state-of-the-art physically based, process driven model that estimates the energy, water and carbon fluxes at the land-atmosphere boundary  and uses a variety of observation-derived products describing physical parameters as inputs (Best et al., 2011; Clark et al., 2011). DALEC, on the other hand, is a simplified terrestrial C-cycle model which is calibrated independently at each location retrieving both process parameters and initial conditions using the CARbon DAta MOdel fraMework (CARDAMOM) model-data fusion system. CARDAMOM ingests satellite based remotely sensed estimates of the state of terrestrial ecosystems (Bloom and Williams, 2015; Bloom et al., 2016; Smallman et al., 2017).

[Section 3.1]
Therefore, DALEC has interannual variability in LAI and soil carbon stocks and can adjust the parameters to find the most likely estimates in combination with other data, whereas these parameters remain constant in JULES. This is potentially advantageous for DALEC, although the use of a climatology in JULES means that noise in the MODIS LAI estimates will be averaged out.

Reviewer comment:
Figure 6 can be improved by adding two additional subfigures that show the difference between DALEC and JULES posterior fluxes. Also the addition of maps showing the related uncertainty in posterior NEE would be helpful.

Author response:
Figure 6 (Fig. 8 in response) has been adapted to show the difference between DALEC and JULES. We don't think that the maps of posterior uncertainty add much beyond the uncertainties shown on the monthly fluxes in Fig. 5.

Reviewer comment:
P.2, the abbreviation GHG is used before it is introduced.

Author response:
Done

Reviewer comment:
P.5, please include an indication of the uncertainty related to the CO2 measurements

Author response:
Done – added to Data selection and model uncertainty section (2.2.1)

Change to manuscript:
Monthly average measurement uncertainty is around 0.9 ppm.

Reviewer comment:
P.10, please mention the spatiotemporal resolution of the prior ocean flux estimates.

Author response:
This was already mentioned in Table 3 (now Table 2).

Reviewer comment:
P.11, to reduce initial confusion, it might be useful to move the paragraph starting at line 350 (about the variable time instead of space dimension) to before equation 4.

Author response:
We think moving this paragraph to before the equation would move the equation too far away from the description of its key elements however, in the paragraph you mention, we have clarified that k relates to equation 4.

Change to manuscript:
Therefore, in this case k in Eq. 4 is more specifically the unknown number of time periods resolved in the inversion, which is important because CO2 fluxes vary strongly in time and have high uncertainty in their temporal variation.

Reviewer comment:
Please be consistent in the colors you use: e.g. the colors for DALEC and JULES are reversed between figures 2 and 5.

Author response:
Done

Reviewer comment:
The section numbering in chapter 4 should be updated, now 4.2 comes after 4.3

Author response:
Done

[revised manuscript text omitted]
 12, 24, 48 and 72 hours back in time, as well as an inversion using integrated footprints combined with monthly fluxes. DALEC NEE was used as the prior flux in this test.

| | |
|---|---|
| **Prior** | − 355 |
| **Posterior** | |
| Integrated footprints | $79\pm^{103}_{106}$ |
| 12-hour back footprints | $-207\pm^{85}_{85}$ |
| 24-hour back footprints | $-356\pm^{87}_{88}$ |
| 48-hour back footprints | $-382\pm^{92}_{86}$ |
| 72-hour back footprints | $-412\pm^{110}_{101}$ |

370

375

[Figure]

**Figure S1:** The domain used to calculate NAME footprints. The four edge boxes correspond to four basis functions. The hatched box is the main area of focus for this study and basis functions in this area are based on a fractional map of 6–7 different PFTs (Fig. S6).

[Figure]

[Figure]

Data

Modelled mole fractions w/ 24 hour bac

**Figure S2:** Forward modelled mole fractions at Ridge Hill and Tacolneston for part of June 2014 using DALEC NEE fluxes and NAME footprints that are disaggregated back in time for 6, 12, 24, 48 and 72 hours, as well as using integrated footprints with monthly fluxes. Anthropogenic and ocean fluxes have been forward modelled and removed from the data. Shading on the data represents ± **1σ**.

1415

[Figure]

1420  **Figure S3:** Data filtered out in 2014 using the "local-lapse" filter. Left hand bar charts for each site show the average percentage of data removed for each 2-hour period in the day. Right hand bar charts for each site show the number of data points used in the inversion for each month (orange bars) and the number of data points removed prior to the inversion for each month (blue bars).

1425

[Figure]

[Figure]

**Figure S4:** A comparison of the results of three different inversions for 2014 using DALEC prior GPP and TER fluxes and three differently filtered data sets. Local-lapse: the filter used on the final results, a combination of localness and vertical temperature profile metrics. Local-lapse 10am – 4pm: data is filtered with the "local-lapse" filter and then only times between 10am and 4pm are selected. 10am – 4pm: all data between 10am and 4pm is used. Shading represents $5^{th} – 95^{th}$ percentile.

[Figure]

[Figure]

**Figure S5:** Synthetic test results. Synthetic data was produced using DALEC biospheric fluxes and NAME simulations– the "true" flux. Prior fluxes are provided by JULES. The "NEE inversion" only scales NEE in the inversion. The "GPP+TER inversion" scales GPP and TER separately in the inversion. NEE prior PDF ($x_{NEE}$) has Gaussian uncertainty distribution and its standard deviation hyper-parameter ($\sigma_{x_{NEE}}$) has a uniform distribution with a range reflecting an absolute uncertainty of approximately 40–400 Tg (see Table 3 for the comparable set-up for the separate GPP and TER inversion). (a) shows prior and posterior monthly flux estimates for the UK in 2014 compared to the "true" flux. Shading represents the $5^{th}$ – $95^{th}$ percentiles. (b) shows average diurnal cycle in June 2014 for prior and posterior NEE in both inversions, as well as the "true" NEE. (c) shows average diurnal cycle in June 2014 for prior and posterior GPP and TER in the "GPP+TER" inversion, as well as the "true" GPP.

[Figure]

Figure S6

[Figure]

**Figure S6:** Maps of plant functional type (PFT) fraction for each of the 6 PFTs used as spatial basis functions within the sub-domain. Note the scale is logarithmic.

475

480

[Figure]

Posterior flux (kg m⁻² yr⁻¹)

485  **Figure S7:** Posterior TER and GPP flux maps averaged over winter 2013 (December 2013, January – February 2014). (a) Winter TER flux from DALEC inversion. (b) Winter TER flux from JULES inversion. (c) Winter GPP flux from DALEC inversion. (d) Winter TER flux from JULES inversion.

[Figure]

Posterior flux (kg m⁻² yr⁻¹)

**Figure S8:** Posterior TER and GPP flux maps averaged over summer 2014 (June – August 2014). (a) Summer TER flux from DALEC inversion. (b) Summer TER flux from JULES inversion. (c) Summer GPP flux from DALEC inversion. (d) Summer GPP flux from JULES inversion.

495

[Figure]

[Figure]

**Figure S9:** Posterior UK fluxes in 2014. (a-c) Comparison of monthly fluxes and minimum and maximum daily values for TER, GPP and NEE respectively resulting from JULES inversion (blue) and DALEC inversion (orange). (d) Annual $CO_2$ fluxes for TER, GPP and NEE for 2013 and 2014 from DALEC and JULES inversions. Dark bars denote prior annual fluxes, light bars denote posterior annual fluxes. Uncertainty bars represent $5^{th} - 95^{th}$ percentile.

500

[Figure]

[Figure]

Figure **S10:** Annual UK NEE flux estimates from DALEC and JULES inversions for 2013 and 2014. Left bars are prior NEE estimates, right bars are posterior NEE estimates. Dashed bars on the posterior estimates represent annual NEE fluxes for inversions that use fixed anthropogenic fluxes multiplied by $\pm 10\%$. Uncertainty bars represent $5^{th} - 95^{th}$ percentile. Solid uncertainty bars on posterior estimates are the uncertainty on the inversions using normal anthropogenic fluxes. Whiskers on the posterior estimates are the uncertainty on the inversions using anthropogenic fluxes multiplied by $\pm 10\%$.

**Moved (insertion) [7]**

**Moved (insertion) [8]**

[Figure]

530

**Figure S11:** Mean and maximum number of temporal regions each month, taken across the number of algorithm iterations, for each source and spatial region.

535

[Figure]

**Figure S12**: Left: Residual mole fractions for prior and posterior modelled $CO_2$ concentrations in 2014 using DALEC prior biospheric fluxes. Right: Histogram of prior residuals (orange) and posterior residuals (blue). The mean of the histogram represents the mean bias.

1540

[Figure]

[Figure]

Observations – prior    ● Observations – posterior    Number of observations

**Figure S13**: Left: Residual mole fractions for prior and posterior modelled $CO_2$ concentrations in 2014 using JULES prior biospheric fluxes. Right: Histogram of prior residuals (orange) and posterior residuals (blue). The mean of the histogram represents the mean bias.

[Figure]

[Figure]

**Moved up [7]:** Left bars are prior NEE estimates, right bars are posterior NEE estimates. Dashed bars on the posterior estimates represent annual NEE fluxes for inversions that use fixed anthropogenic fluxes multiplied by ±

**Figure S11:** Annual UK NEE flux estimates from CARDAMOM and JULES inversions for 2013 and 2014.

**Moved up [8]:** Uncertainty bars represent $5^{th} - 95^{th}$ percentile. Solid uncertainty bars on posterior estimates are the uncertainty on the inversions using normal anthropogenic fluxes. Whiskers on the posterior estimates are the uncertainty on the inversions using anthropogenic fluxes multiplied by ±

555

**Figure S14:** Left: Residual mole fractions for modelled $CO_2$ concentrations at Weybourne in 2013 using prior DALEC and JULES fluxes, and posterior DALEC and JULES fluxes from both the gross (scaling GPP and TER separately) and net (scaling just NEE) flux inversions. Weybourne data was not included in the inversions. Right: Histogram of residuals. The mean of the histogram represents the mean bias.

[Figure]

**Figure S15**: Posterior monthly net UK $CO_2$ flux (+ve is emission to atmosphere) for the inversion that scales only NEE rather than GPP and TER separately. Orange and blue monthly fluxes are posterior net biospheric (NEE) fluxes for DALEC and JULES respectively. Prior biosphere fluxes from DALEC and JULES are shown in dashed orange and blue lines respectively. Shading represents $5^{th} - 95^{th}$ percentile. The bar charts represent annual net UK $CO_2$ flux for 2013 (left) and 2014 (right). Hashed bars denote prior annual fluxes, solid bars denote posterior annual fluxes. The bar colours correspond to the line colours: left hand bars for each model are NEE fluxes, right hand bars for each model are total fluxes (NEE + fixed sources). Uncertainty bars represent $5^{th} - 95^{th}$ percentile. DA – DALEC. JU – JULES. NEE prior PDF ($x_{NEE}$) has Gaussian uncertainty distribution and its standard deviation hyper-parameter ($\sigma_{x_{NEE}}$) has a uniform distribution with a range reflecting an absolute uncertainty of approximately 40–400 Tg (see Table 3 for the comparable set-up for the separate GPP and TER inversion).

| | | |
|---|---|---|
| **Page 9: [1] Deleted** | **Emily White** | **2/13/19 11:20:00 PM** |

| | | |
|---|---|---|
| **Page 30: [2] Deleted** | **Emily White** | **2/13/19 11:20:00 PM** |
| **Page 34: [3] Deleted** | **Emily White** | **2/13/19 11:20:00 PM** |
| **Page 39: [4] Deleted** | **Emily White** | **2/13/19 11:20:00 PM** |
| **Page 39: [5] Deleted** | **Emily White** | **2/13/19 11:20:00 PM** |
| **Page 39: [6] Deleted** | **Emily White** | **2/13/19 11:20:00 PM** |
| **Page 39: [7] Deleted** | **Emily White** | **2/13/19 11:20:00 PM** |
| **Page 52: [8] Deleted** | **Emily White** | **2/13/19 11:20:00 PM** |